# VisionPulse: Dynamic Visual Sparsity for Efficient Multimodal Reasoning

**Hengbo Xu**[1]  **Shengjie Jin**[1]  **Yanbiao Ma**[1]  **Zhiwu Lu**[1]

## Abstract

With the rapid advancement of large multimodal models (LMMs), inference-time overhead has become a key bottleneck for real-world deployment. Existing methods typically prune visual tokens at prefill, assuming the required visual evidence remains static during reasoning. However, we empirically show that visual evidence is strongly step-dependent: only a sparse subset of visual tokens is critical at each decoding step, and the critical set evolves across reasoning. Furthermore, we identify a coupled bottleneck where redundant visual context can steer the model toward query-irrelevant regions, lengthening the reasoning trace. Guided by these insights, we propose **VisionPulse**, a step-wise visual token pruning framework during reasoning. VisionPulse computes a lightweight visual attention mass to estimate the step-wise retention budget by exploiting its strong positive correlation with LMMs' effective visual token usage and retain only the most critical tokens under this budget. By enforcing visual sparsity during reasoning, VisionPulse filters redundant visual context while preserving relevant visual evidence, shortening reasoning traces naturally. Extensive experiments show that VisionPulse only retains 5% of visual tokens per step with reasoning traces shortened by 11.2%, while keeping accuracy almost unchanged.

## 1. Introduction

Large multimodal models (LMMs) (Jaech et al., 2024; Bai et al., 2025; Wang et al., 2025c) have recently achieved substantial progress in multimodal reasoning, enabling vision-centric, multi-step chain-of-thought (CoT) reasoning (Wang et al., 2025a; Yang et al., 2025d; Zhou et al., 2025; Wang et al., 2025d; Sun et al., 2025; Dai et al., 2026) across tasks such as scientific problem solving, chart understanding, and embodied perception. However, as reasoning capabilities continue to improve, inference-time overhead has emerged as a primary bottleneck for real-world deployment (Kwon et al., 2023; Lin et al., 2024).

Under existing multimodal decoding paradigms, most LMMs (Li et al., 2024; Bai et al., 2025; Wang et al., 2025c; Yao et al., 2024) treat image and text tokens uniformly within a shared self-attention mechanism, where each decoding step computes interactions over the entire multimodal context. On the visual side, the strong performance of LMMs depends on fine-grained visual representations. As visual understanding demands grow, images and videos are represented more densely, resulting in substantially more visual tokens and higher computational cost. On the reasoning side, as LMMs become stronger multimodal reasoners, they are increasingly able to tackle complex, out-of-distribution tasks. Meanwhile, when faced with hard queries, LMMs typically generate longer multi-step reasoning traces, which expand the language context during decoding. Consequently, each decoding step must attend to both an expanding language context and a large visual token budget, leading to a sharp increase in inference overhead.

A natural question thus arises in multimodal reasoning: are all visual tokens and generated reasoning tokens equally necessary throughout the entire reasoning process? Prior studies (Wang et al., 2021; Liang et al., 2022; Yu et al., 2022) have noted that the information content in images is often sparse. Moreover, for a given query, only a limited subset of visual information is actually relevant, motivating a range of visual compression methods for efficient multimodal inference (Yang et al., 2025a; Chen et al., 2024b; Zhang et al., 2025d;a; Yang et al., 2025c). These methods typically perform compression at the prefill stage by selecting a fixed subset of visual tokens for subsequent decoding. While effective for direct-answer generation, this design implicitly assumes that the selected visual tokens remain relevant and unchanged throughout the reasoning process.

To examine this assumption for multimodal reasoning, we conduct an empirical analysis of multimodal reasoning models and focus on how visual tokens are utilized. Specifically, we quantify the aggregate attention allocated to visual to-

---

[1]Gaoling School of Artificial Intelligence, Renmin University of China, Beijing, China. Correspondence to: Zhiwu Lu <luzhiwu@ruc.edu.cn>.

*Proceedings of the 43rd International Conference on Machine Learning*, Seoul, South Korea. PMLR 306, 2026. Copyright 2026 by the author(s).

(a)

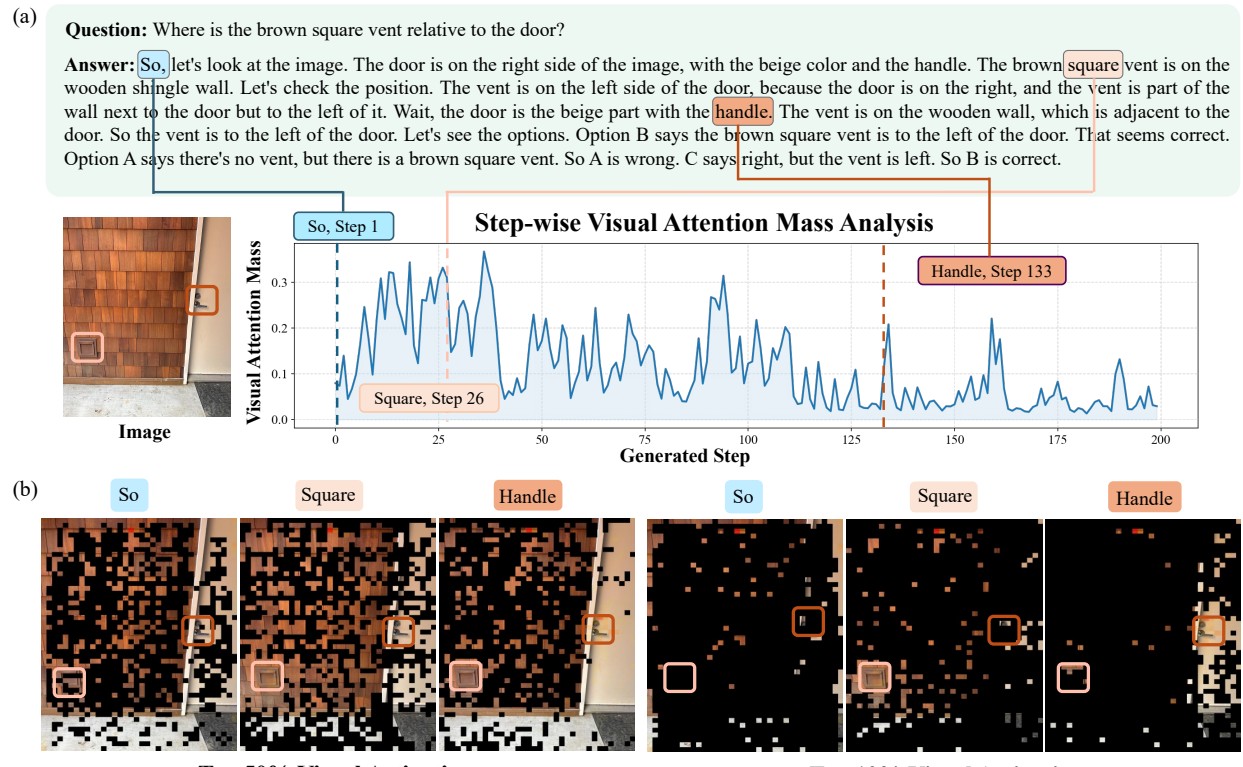

**Question:** Where is the brown square vent relative to the door?

**Answer:** So, let's look at the image. The door is on the right side of the image, with the beige color and the handle. The brown square vent is on the wooden shingle wall. Let's check the position. The vent is on the left side of the door, because the door is on the right, and the vent is part of the wall next to the door but to the left of it. Wait, the door is the beige part with the handle. The vent is on the wooden wall, which is adjacent to the door. So the vent is to the left of the door. Let's see the options. Option B says the brown square vent is to the left of the door. That seems correct. Option A says there's no vent, but there is a brown square vent. So A is wrong. C says right, but the vent is left. So B is correct.

**Figure 1.** **Dynamic visual activations during multimodal reasoning. (a) Step-wise Visual attention mass over decoding.** We measure the step-wise visual attention mass (total attention allocated to visual tokens) during reasoning. Visual evidence is strongly step-dependent: it remains negligible in text-dominated steps, but increases when the reasoning involves referenced entities (e.g., *square*, *handle*). **(b) Visual attention heatmap at representative steps.** Top-50% and top-10% visual-attention activation masks show that attention concentrates on regions corresponding to referenced entities at high-mass steps, but remains diffuse at low-mass steps. This indicates that the amount and the subset of critical visual tokens vary during reasoning.

kens at each decoding step (i.e., visual attention mass) and analyze the corresponding attention heatmaps for representative decoding steps. As shown in Figure 1, the need for visual evidence is activated on demand as the reasoning state evolves. It is negligible in text-dominated steps but increases substantially when the model must ground specific visual entities. Moreover, LMM's visual attention shifts across regions over time, indicating that the critical visual tokens vary during reasoning. Therefore, no single fixed subset can remain optimal throughout reasoning.

These observations thus expose a fundamental mismatch in existing static pruning for efficient multimodal reasoning. Most prior methods fix a single visual subset once at the start of decoding, when the model generates the first token and allocates little attention mass to visual tokens, as shown in Figure 1. As a result, a fixed subset is inevitably misaligned with later demands: it may discard visual tokens that become crucial at subsequent steps that require visual evidence, while still preserving redundant context during text-dominated reasoning. This mismatch becomes particularly pronounced under aggressive retention, where early pruning errors cannot be corrected and may propagate through the entire reasoning process.

Motivated by these observations, we propose **VisionPulse**, a **training-free** dynamic visual pruning framework for multimodal reasoning. Unlike prior methods (Chen et al., 2024b; Zhang et al., 2025d; Xing et al., 2024) that prune visual tokens once at prefill, VisionPulse performs step-wise visual token selection during reasoning, enabling the model to select critical visual tokens as the reasoning state evolves. Beyond preserving accuracy, we uncover a coupled bottleneck: retaining redundant visual context at every step steers LMMs toward query-irrelevant visual cues, yielding unnecessary and even sometimes harmful reasoning traces (see Section 5.3). Furthermore, we observe a strong positive correlation between visual attention mass and LMMs' effective visual activation, which motivates a simple yet effective visual mass guided budgeting strategy. Specifically, Vision-Pulse uses this lightweight mass signal to set a per-step retention budget and retains only the most critical visual tokens under this budget. Extensive experiments on seven benchmarks demonstrate that VisionPulse achieves performance comparable to the full-token baseline under extreme visual token retention settings (10% and 5%), while reducing unnecessary generation length by 12.3% and 11.2%, substantially outperforming existing methods.

Our contributions are summarized as follows: **1)** We empirically show that the need for visual evidence in multimodal reasoning is highly step-dependent, and the critical visual token set evolves over decoding steps. **2)** We identify a coupled bottleneck in multimodal reasoning, where redundant visual context can steer LMMs to query-irrelevant regions and lengthen the reasoning trace. **3)** We propose **VisionPulse**, a training-free dynamic visual token pruning framework that selects critical visual tokens at each decoding step via a lightweight visual mass signal. **4)** Extensive experiments on seven benchmarks validate that VisionPulse mitigates redundant reasoning traces, improving efficiency while keeping accuracy almost unchanged.

## 2. Related Work

**Multimodal Reasoning**  Large Reasoning Models (Guo et al., 2025; Jaech et al., 2024; Team et al., 2025) improve performance by explicitly generating chain-of-thought (CoT), but stronger reasoning typically requires longer generated sequences and thus higher inference cost, especially on difficult problems (Muennighoff et al., 2025; Snell et al., 2024; Yeo et al., 2025). Efficient reasoning methods (Choi et al., 2025; Xia et al., 2025; Zhang et al., 2025b; Xiao et al., 2025; Yang et al., 2025b) therefore primary focus on more concise CoT generation while largely ignoring the contribution of visual tokens to inference cost. In contrast, we first study multimodal reasoning efficiency from a unified perspective and show that redundant visual context is doubly costly: it increases per-step attention computation and can also lengthen the reasoning trace.

**Visual Compression**  Visual token redundancy is prevalent in LMMs: a single high-resolution image can yield thousands of tokens, and videos can scale to tens of thousands (Wang et al., 2025b; Wu et al., 2024). Existing pruning methods broadly fall into two categories: (1) compressing visual tokens using attention patterns from the vision encoder (Yang et al., 2025a;c; Shang et al., 2025; Arif et al., 2025; Liu et al., 2025), and (2) selecting a query-relevant subset at prefill (Chen et al., 2024b; Zhang et al., 2025d; Khaki et al., 2025; Zhang et al., 2025c). Despite these differences, most methods compress only once before decoding, implicitly assuming the retained subset remains sufficient throughout reasoning. In contrast, our analysis shows that visual reliance varies substantially across decoding steps, which directly motivates the design of VisionPulse.

## 3. Motivation

### 3.1. Inference Cost Analysis of Multimodal Reasoning

In multimodal reasoning, given a test query $q = \{p, v\}$, where $p$ is the text prompt and $v$ is the visual input (images or videos), LMMs integrate visual and textual information

to generate an answer through multi-step reasoning. Under the standard KV-cache implementation, the model first processes the full input once to initialize the KV-cache (prefill stage), and then generates $g$ tokens autoregressively while updating the KV-cache (decoding stage). We decompose the total floating-point operations (FLOPs) as follows:

$$
\begin{aligned}
\mathcal{F}_{\text{total}} &= \mathcal{F}_{\text{pre}} + \sum_{t=1}^{g} \mathcal{F}_{\text{dec}} \\
&\approx L \cdot \Big[ \underbrace{(p+v)(8d^2 + 4md) + 4d(p+v)^2}_{\text{Prefill Stage}} \Big] \\
&\quad + L \cdot \underbrace{\sum_{t=1}^{g} \Big[ (8d^2 + 4md) + 4d(p+v+t) \Big]}_{\text{Decoding Stage}},
\end{aligned} \tag{1}
$$

where $L$ denotes the number of layers, $d$ is the hidden dimension, $m$ is the FFN intermediate size, and $g$ is the number of generated tokens. Eq. 1 highlights a coupling inference burden between the reasoning length $g$ and the initial multimodal context length $(p+v)$. Unlike prefill which is performed once, decoding applies self-attention over a context growing from $(p+v)$ to $(p+v+g)$, the total cost scales as $O(g(p+v) + g^2)$. In multimodal settings where the visual context often dominates the input length $(v \gg p)$, visual tokens introduce a large baseline cost per decoding step. As $g$ increase, this results in significantly higher marginal costs compared to text-only models, making the visual context becoming a dominant factor in total end-to-end latency.

### 3.2. Dynamic Visual Attention in Multimodal Reasoning

Given the redundancy in visual tokens, visual compression has emerged as a promising strategy for enhancing efficiency. Most existing methods adopt a static selection paradigm: they estimate token importance once at prefill and keep this fixed subset throughout decoding. This implicitly assumes that the visual evidence required for reasoning remains unchanged over time. We argue that this assumption is mismatched to multimodal reasoning, where the model's reliance on visual evidence changes as the reasoning state evolves. To validate this, we visualize visual attention in Qwen3-VL-4B-Thinking during CoT reasoning as shown in Figure 1. Specifically, we quantify step-wise visual attention mass and examine visual attention heatmaps at representative steps, leading to three key observations.

First, visual reliance is strongly step-dependent rather than constant. As shown in Figure 1a, visual attention activates when LMM confirms visual details (such as *handle*), while diminishing significantly during steps dominated by textual reasoning (e.g., *"Let's check the options"*). It indicate that visual evidence is dynamic and activated on demand as the reasoning state evolves. Second, the set of critical visual tokens varies during reasoning. When visual attention mass

increases, the activated regions also shift with the current reasoning content. In Figure 1b, when the model generates entity-specific tokens (e.g., *handle*), attention explicitly concentrates on the corresponding region. In contrast, during prefill, the model focuses on generating the initial token (e.g., *so* in step 1), providing little semantic guidance for identifying the relevant regions. Consequently, selecting a single fixed visual subset at prefill is suboptimal: it may discard tokens that become important later, while retaining irrelevant context during low-reliance steps.

Third and critically, we identify a coupled bottleneck in multimodal reasoning. Maintaining the full visual context throughout decoding can bias the model toward generating unnecessary visual cues that contribute little to the reasoning process. For instance, in Figure 1, describing *handle* is irrelevant for judging the spatial relation between the door and the brown square. Such redundancy not only incurs avoidable computational overhead but also introduces visual noise that undermines the reliability of the reasoning. We provide a more comprehensive discussion of this phenomenon in Section 5.3. Overall, these observations motivate a dynamic pruning mechanism that aligns visual budgets with evolving reasoning demands, maximizing compression efficiency without compromising LMMs' reasoning performance.

# 4. VisionPulse

## 4.1. Overview Architecture

To mitigate the decoding-time bottleneck from redundant visual context, we propose **VisionPulse**, a **training-free** method for step-wise visual token pruning in multimodal reasoning. As shown in Figure 2, VisionPulse keeps only the most critical visual tokens at each step, enabling step-dependent selection during decoding. Furthermore, it uses a lightweight visual attention mass signal $M_{\text{vis}}^t$ to determine the step-wise retention budget $K_t$, retaining the top-$K_t$ visual tokens to form the pruned decoding context. We describe the details of each module below.

## 4.2. Step-wise Visual Token Pruning

In multimodal reasoning, LMMs exhibit highly step-dependent visual dependencies during decoding rather than maintaining a constant focus. As a result, retaining the complete visual context at every decoding step imposes an unnecessary computation burden. Motivated by this, we adopt a step-wise pruning paradigm that performs adaptive pruning at every decoding step.

Formally, let the visual token set be $X_v = \{v_1, v_2, ..., v_N\}$. At each decoding step $t$, given the current query token $q_t$, our objective is to identify a critical subset $X_v^t \subseteq X_v$ and drop the remaining tokens. This fundamentally differs from static pruning methods, which estimate visual token importance

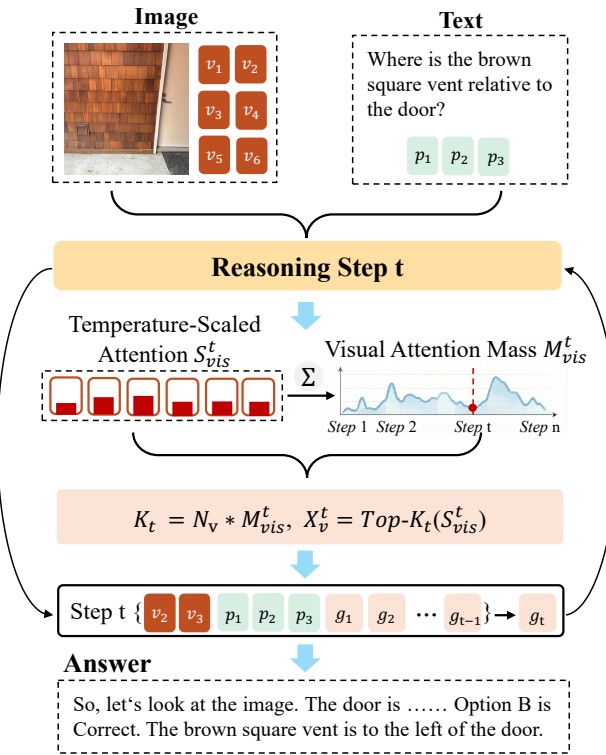

*Figure 2.* **Overview of VisionPulse.** At decoding step $t$, Vision-Pulse adaptively prunes visual tokens by computing a lightweight visual attention mass $M_{\text{vis}}^t$ to determine the step-wise budget $K_t$, and retains the top-$K_t$ tokens for decoding. Temperature scaling is used to enable adjustable compression ratios.

once at prefill and keep a fixed subset throughout decoding. In contrast, we re-estimate token importance at every step to track the evolving reasoning focus. For simplicity, we follow FastV (Chen et al., 2024b) for importance estimation but recompute it at every decoding step $t$. Specifically, we compute the importance scores at layer $l_a$ and start pruning from $l_a$. At the anchor layer $l_a$, we aggregate attention probabilities over all heads to define the significance score for each visual token:

$$S_i^t = \frac{1}{H} \sum_{h=1}^{H} A_{t,h}^{(l_a)}(q_t, v_i), \qquad (2)$$

where $A_{t,h}^{(l_a)}(q_t, v_i)$ denotes the attention probability from the current query $q_t$ to the visual key $v_i$ at head $h$ of layer $l_a$. This importance estimator is flexible and can be replaced by alternative scoring functions used in prior work (Xing et al., 2024; Yin et al., 2025). Notably, given $\{S_i^t\}_{i=1}^N$, instead of using a fixed retention ratio, we determine a step-specific retention budget $K_t$ and retain Top-$K_t$ visual tokens:

$$X_v^t = \{v_i \mid i \in \text{Top-}K_t(\{S_i^t\}_{i=1}^N)\}. \qquad (3)$$

## 4.3. Visual Mass Guided Dynamic Budget

A key challenge in step-wise visual token pruning is that a fixed retention ratio cannot accommodate the heterogeneous

visual needs across decoding steps. Some steps require broad visual evidence, while others are largely language-driven and only need minimal visual support. Furthermore, a larger visual attention does not necessarily imply a broader visual coverage: it could either concentrate on a few salient visual tokens or spread over many tokens. To motivate a step-wise budget, we first conduct an empirical analysis across decoding steps and find that the visual attention mass $M_{\text{vis}}^t$—a scalar measuring how much attention probability is allocated to visual tokens at step $t$—is strongly correlated with the number of effectively activated visual tokens.

Concretely, to characterize how many visual tokens receive non-negligible attention, we define the active visual token count under a threshold $\delta$ as:

$$N_{\text{act}}^t(\delta) = \left| \left\{ i \in \{1, \ldots, N_v\} \mid \bar{A}_t^{(l_a)}(q_t, v_i) > \delta \right\} \right|, \quad (4)$$

where $N_v = |X_v|$ is the number of visual tokens. Meanwhile, we quantify the overall visual dependency at decoding step $t$ by the visual attention mass:

$$m_{t,h}^{\text{vis}} = \sum_{i=1}^{N_v} A_{t,h}^{(l_a)}(q_t, v_i), \quad M_{\text{vis}}^t = \frac{1}{H} \sum_{h=1}^{H} m_{t,h}^{\text{vis}}. \quad (5)$$

As shown in Figure 3, $M_{\text{vis}}^t$ exhibits a strong positive linear correlation with $N_{\text{act}}^t(\delta)$ across decoding steps, and the trend remains consistent across a range of thresholds (Pearson $r$ ranges from 0.82 to 0.95). This suggests that higher visual dependency tends to lift more tokens above non-negligible attention levels, instead of only amplifying attention on a few tokens.

Motivated by this observation, we use the visual attention mass as a lightweight predictor to allocate a step-wise retention budget. At decoding step $t$, we estimate $M_{\text{vis}}^t$ at the anchor layer $l_a$ using a temperature-scaled attention distribution with $\tau < 1$, which enables controllable pruning ratios and suppresses the softmax long tail over $X_v$. Specifically, we compute the temperature-scaled attention probability:

$$A_{t,h}^{(l_a,\tau)}(q_t, v_i) = \frac{\exp\left(Z_{t,h}^{(l_a)}(q_t, v_i)/\tau\right)}{\sum_j \exp\left(Z_{t,h}^{(l_a)}(q_t, j)/\tau\right)}, \quad (6)$$

where $Z_{t,h}^{(l_a)}(q_t, v_i)$ denotes the pre-softmax attention logit and $\tau$ is the temperature. Using $A_{t,h}^{(l_a,\tau)}$, we compute the visual attention mass $M_{\text{vis}}^t$ following Equation (5). For budgeting, we adopt the head-max variant $M_{vis,max}^t$ when computing $K_t$, yielding a conservative estimate that avoids under-budgeting visual-dominant heads (Kang et al., 2025; Wan et al., 2025). Finally, we predict the pruning budget directly from $M_{\text{vis,max}}^t$:

$$K_t = M_{\text{vis,max}}^t N_v. \quad (7)$$

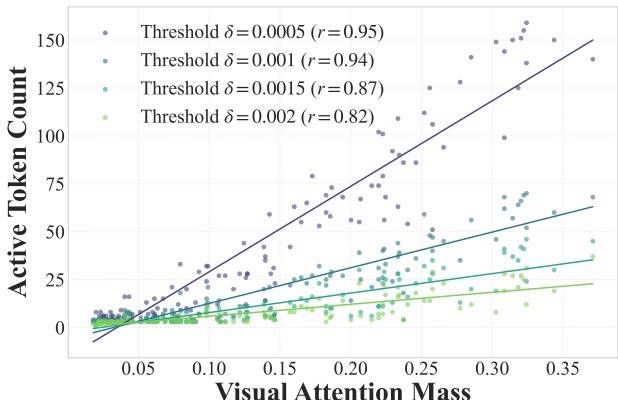

*Figure 3.* **Visual attention mass predicts the number of activated visual tokens.** Scatter plot of visual attention mass $M_{\text{vis}}^t$ v.s. active visual token count $N_{\text{act}}^t(\delta)$ under different activation thresholds $\delta$. Across thresholds, $M_{\text{vis}}^t$ shows a strong positive linear correlation with $N_{\text{act}}^t(\delta)$, supporting dynamic budget allocation.

## 5. Experiments

### 5.1. Experimental Setup

To evaluate our method on general multimodal reasoning, we conduct experiments on seven widely adopted benchmarks (Masry et al., 2022; Yu et al., 2024; xAI, 2024; Qian et al., 2025; Wang et al., 2024; Mathew et al., 2022; Chen et al., 2024a). We compare our approach with three representative state-of-the-art visual token pruning methods: VisionZip (Yang et al., 2025a), FastV (Chen et al., 2024b), and LOOK-M (Wan et al., 2024). VisionZip selects visual tokens according to the visual attention extracted from the last layer of the visual encoder, and performs pruning before the prefill stage. FastV prunes visual tokens at an intermediate layer of the LMM during the prefill stage, where $l_a = 17$. LOOK-M performs KV-cache compression across all layers during prefill, preserving important tokens in a layer-wise manner. Despite differing in pruning granularity and application stage, these methods make a single static pruning decision and therefore overlook the step-dependent need for visual evidence during reasoning. For a controlled comparison, we build VisionPulse on top of FastV's importance estimator (FastV-VP) and prune at the same layer.

### 5.2. Main Results

We evaluate all methods under two aggressive settings with ~10% and ~5% visual token retention. For fair comparison, we clamp $K_t$ to match the target average retention: $(0.05N, 0.10N)$ for 10% and $(0.01N, 0.05N)$ for 5%.

The results are shown in Table 1, where our method consistently achieves markedly better performance than existing approaches. Specifically, when using at most 5% visual tokens at each decoding step, our method preserves nearly 100% of the original performance. Even under the more extreme setting where only 1% of visual tokens are retained

*Table 1.* **Main results under aggressive visual-token retention (Qwen3-VL Thinking 4B).** We compare our FastV-VP with prior pruning methods across seven benchmarks. Each cell reports the absolute value (top; $L$ = generated tokens, $A$ = accuracy) and the relative change from the full-token baseline (bottom). Results are grouped by retention levels (retain $\leq$10% and $\leq$5% tokens).

| Method | CharXiv RQ | | InfoVQA | | ChartQA | | MMStar | | RealWorld QA | | MMVet | | MIA-Bench | | Avg | |
|---|---|---|---|---|---|---|---|---|---|---|---|---|---|---|---|---|
| | $L\downarrow$ | $A\uparrow$ | $L\downarrow$ | $A\uparrow$ | $L\downarrow$ | $A\uparrow$ | $L\downarrow$ | $A\uparrow$ | $L\downarrow$ | $A\uparrow$ | $L\downarrow$ | $A\uparrow$ | $L\downarrow$ | $A\uparrow$ | $L\downarrow$ | $A\uparrow$ |
| **Upper Bound (Retain 100% Tokens)** | | | | | | | | | | | | | | | | |
| **Baseline** | 4068.0 | 47.60 | 623.1 | 84.37 | 510.0 | 77.12 | 2058.5 | 72.47 | 678.0 | 72.81 | 1642.0 | 60.96 | 2231.3 | 93.44 | 1687.3 | 72.68 |
| **Retain $\leq$ 10% Tokens** | | | | | | | | | | | | | | | | |
| Visionzip | 4986.2 | 13.90 | 2533.3 | 22.66 | 2039.7 | 30.24 | 2496.8 | 51.53 | 726.0 | 59.08 | 2346.7 | 41.19 | 3089.6 | 88.14 | 2602.6 | 43.82 |
| | +22.6% | -70.8% | +306.6% | -73.1% | +299.9% | -60.8% | +21.3% | -28.9% | +7.1% | -18.9% | +42.9% | -32.4% | +38.5% | -5.7% | +54.2% | -39.7% |
| FastV | 5960.1 | 12.70 | 2963.6 | 20.63 | 1485.5 | 16.28 | 2275.3 | 50.40 | 575.8 | 55.95 | 3296.8 | 28.12 | 2714.9 | 82.53 | 2753.1 | 38.09 |
| | +46.5% | -73.3% | +375.6% | -75.5% | +191.3% | -78.9% | +10.5% | -30.5% | -15.1% | -23.2% | +100.8% | -53.9% | +21.7% | -11.7% | +63.2% | -47.6% |
| LOOK-M | 5555.2 | 19.80 | 2694.1 | 40.94 | 2007.1 | 57.68 | 2090.6 | 58.87 | 481.8 | 67.84 | 2518.9 | 50.05 | 3376.1 | 88.71 | 2674.8 | 54.84 |
| | +36.6% | -58.4% | +332.4% | -51.5% | +293.5% | -25.2% | +1.6% | -18.8% | -28.9% | -6.8% | +53.4% | -17.9% | +51.3% | -5.1% | +58.5% | -24.5% |
| **FastV-VP (Ours)** | 3770.7 | 47.30 | 530.7 | 83.62 | 422.9 | 76.72 | 1873.7 | 72.20 | 564.0 | 73.20 | 1329.8 | 61.79 | 1870.8 | 93.99 | 1480.4 | 72.69 |
| | -7.3% | -0.6% | -14.8% | -0.9% | -17.1% | -0.5% | -9.0% | -0.4% | -16.8% | +0.5% | -19.0% | +1.4% | -16.2% | +0.6% | -12.3% | +0.0% |
| **Retain $\leq$ 5% Tokens** | | | | | | | | | | | | | | | | |
| Visionzip | 3751.7 | 12.30 | 1468.5 | 22.04 | 957.3 | 19.43 | 2289.6 | 44.93 | 671.2 | 52.02 | 1918.1 | 36.56 | 2914.1 | 87.88 | 1995.8 | 39.31 |
| | -7.8% | -74.2% | +135.7% | -73.9% | +87.7% | -74.8% | +11.2% | -38.0% | -1.0% | -28.6% | +16.8% | -40.0% | +30.6% | -6.0% | +18.3% | -45.9% |
| FastV | 4259.7 | 11.60 | 2239.7 | 21.78 | 1207.3 | 12.88 | 2040.4 | 50.13 | 613.3 | 54.12 | 2606.5 | 24.27 | 2342.2 | 75.03 | 2187.0 | 35.69 |
| | +4.7% | -75.6% | +259.4% | -74.2% | +136.7% | -83.3% | -0.9% | -30.8% | -9.5% | -25.7% | +58.7% | -60.2% | +5.0% | -19.7% | +29.6% | -50.9% |
| LOOK-M | 6448.0 | 17.20 | 5511.1 | 27.39 | 2577.2 | 28.80 | 2265.4 | 54.87 | 573.0 | 68.37 | 2859.2 | 35.05 | 4329.3 | 80.66 | 3509.0 | 44.62 |
| | +58.5% | -63.9% | +784.5% | -67.5% | +405.3% | -62.7% | +10.1% | -24.3% | -15.5% | -6.1% | +74.1% | -42.5% | +94.0% | -13.7% | +108.0% | -38.6% |
| **FastV-VP (Ours)** | 3645.1 | 45.20 | 665.0 | 81.90 | 510.0 | 75.16 | 1875.4 | 70.53 | 538.0 | 72.54 | 1456.2 | 59.00 | 1800.1 | 95.09 | 1498.5 | 71.35 |
| | -10.4% | -5.0% | +6.7% | -2.9% | +0.0% | -2.5% | -8.9% | -2.7% | -20.6% | -0.4% | -11.3% | -3.2% | -19.3% | +1.8% | -11.2% | -1.8% |

per step, the model still maintains 98.2% of the full-token performance. Moreover, unlike prior methods, we significantly shorten the reasoning trajectory while keeping the performance nearly unchanged. Compared to the full-token baseline, the average generation length is reduced by 12.3% and 11.2% under the two settings, respectively. In contrast, static pruning baselines degrade sharply, with average accuracy dropping by 24.5%–50.9% and generation length increasing by 18.3%–108.0% under aggressive retention. These results indicate that VisionPulse does not only reduce the number of visual tokens, but instead removes truly irrelevant visual information at each decoding step, consistently retaining only the tokens critical to sustain effective visual activation for the current reasoning stage.

Furthermore, we observe that incorrect pruning strategies not only degrade performance but also unexpectedly increase reasoning cost substantially. Static methods (e.g., VisionZip, FastV, and LOOK-M) often remove critical tokens prematurely, forcing the model to compensate via longer reasoning generation, which both reduces accuracy and increases computational overhead. For example, LOOK-M incurs a 108% increase in average generation length, yet its accuracy still drops by 38.6%. In contrast, VisionPulse maintains shorter generations and stable performance even under extremely low visual retention. Even on vision-intensive tasks such as ChartQA, our method remains stable, with only a 2.5% performance drop under 5% visual-token reten-

tion. Overall, these results support our hypothesis that visual dependency varies throughout the reasoning process, and fixed-budget pruning strategies are inherently mismatched to this property. By using visual attention mass as a lightweight predictor, VisionPulse better tracks step-wise visual dependency and enables more efficient pruning.

### 5.3. The Coupled Bottleneck in Multimodal Reasoning

Beyond inference-time overhead, we observe a coupled bottleneck that affects both the length and reliability of multimodal reasoning. During decoding, LMMs remain exposed to abundant visual context even when the current step requires little visual evidence, which can steer the model toward query-irrelevant regions and induce redundant descriptions, thereby elongating the reasoning trace.

As illustrated in Figures 1 and 4, this phenomenon not only lengthens reasoning traces but can also lead the model toward harmful reasoning paths. Specifically, in Figure 1a, the model elaborates on descriptive details (e.g., the handle and color of the door) that are irrelevant to the target reasoning. Furthermore, in Figure 4, an early description of the car biases subsequent inference, prompting the model to infer the traffic light state from the car's motion instead of the signal itself. Such failure modes are unacceptable in safety-critical scenarios such as autonomous driving. Additional examples are provided in the Appendix. In contrast, step-wise

| A. Test Query | B. Baseline: Before Pruning (Visual Noise Interference) | ❌ Reasoning Error |
|---|---|---|

So, let's look at the traffic lights in the image. The traffic lights are the ones hanging over the road, right? In the picture, the traffic lights are showing red, I think. Because the cars are stopped, so the light must be red. Let me check again. The traffic lights are the ones with the orange lights? Wait, no, in the image, the traffic lights are the ones on the overhead gantry. They are lit up, but since the cars are stopped, it's probably red. So the answer would be B. No, because they are not green.

| C. Ours: After Pruning (Focused Reasoning) | ✔ Correct Answer |
|---|---|

So, let's look at the image. The traffic lights are the ones hanging over the road. In the picture, the lights are red, right? Because they're showing red, which means stop. So the question is asking if they're green. Since they're red, the answer should be B. No.

**Question:** Are the traffic lights green?

*Figure 4.* **Coupled bottleneck in multimodal reasoning.** With full visual context (top), redundant visual tokens remain available throughout decoding and can draw attention to query-irrelevant cues, leading to unnecessary descriptions and even erroneous reasoning (e.g., inferring the traffic-light state from vehicles rather than the signal). With step-wise visual pruning (bottom), the model retains only query-relevant visual evidence for the current step, yielding more focused reasoning and correct prediction.

*Table 2.* **Ablation study on Qwen3-VL-Thinking-4B under aggressive retention (≤5%).** We analyze the effect of different budget allocation strategies used for FastV-VP on three benchmarks.

| Method | Avg Ret. | RealWorld QA | | MMVet | | MIA-Bench | | Avg $L\downarrow$ | Avg $A\uparrow$ |
|---|---|---|---|---|---|---|---|---|---|
| | | $L\downarrow$ | $A\uparrow$ | $L\downarrow$ | $A\uparrow$ | $L\downarrow$ | $A\uparrow$ | | |
| **Baseline** | 100% | 678.0 | 72.81 | 1642.0 | 60.96 | 2231.3 | 93.44 | 1517.1 | 75.74 |
| **Retain ≤ 5% Tokens** | | | | | | | | | |
| FastV | 5.0% | 613.3 -9.5% | 54.12 -25.7% | 2606.5 +58.7% | 24.27 -60.2% | 2342.2 +5.0% | 75.03 -19.7% | 1853.9 +22.2% | 51.14 -32.5% |
| FastV-VP (Fix Ratio) | 1.0% | 509.7 -24.8% | 71.90 -1.3% | 2913.4 +77.4% | 49.17 -19.3% | 2398.0 +7.5% | 92.03 -1.5% | 1940.4 +27.9% | 71.03 -6.2% |
| FastV-VP (Fix Ratio) | 5.0% | 678.6 +0.1% | 72.81 +0.0% | 1557.7 -5.1% | 59.45 -2.5% | 1971.0 -11.7% | 93.22 -0.2% | 1402.4 -7.6% | 75.16 -0.8% |
| FastV-VP (Random Budget) | 3.0% | 519.0 -23.5% | 69.28 -4.9% | 1820.9 +10.9% | 58.02 -4.8% | 2219.5 -0.5% | 91.49 -2.1% | 1519.8 +0.2% | 72.93 -3.7% |
| **FastV-VP (Dynamic Budget)** | 1.9% | 538.0 -20.6% | 72.54 -0.4% | 1456.2 -11.3% | 59.00 -3.2% | 1800.1 -19.3% | 95.09 +1.8% | **1264.8 -16.6%** | **75.54 -0.3%** |

visual sparsification retains only the evidence required at each step, reducing redundant generations while preserving correctness. These observations support our claim that step-wise pruning paradigm better aligns the visual budget with evolving visual reliance during reasoning.

### 5.4. Ablation Study

We conduct ablation studies to investigate the contribution of each component in VisionPulse. As shown in Table 2, both components of our method are effective and contribute to the overall gains.

(1) **Step-wise pruning is necessary:** Under low retention, aligning pruning with the step-wise need for visual evidence is critical for both efficiency and correctness. Using FastV as a static baseline, fixed 5% retention leads to longer generations (+22.2%) and a large accuracy drop (-32.5%) in Table 2. This suggests that prefill-only pruning removes evidence required later, forcing the LMM to compensate with longer reasoning yet still failing to answer correctly. (2) **Dynamic budgeting improves the trade-off:** within

FastV-VP, a fixed 1% budget causes a 6.2% accuracy drop with 27.9% longer generations, while a fixed 5% budget cannot adapt to step-wise variation and leaves redundant context in low-demand steps. In contrast, dynamic budgeting achieves a better balance, reducing length by 16.6% with negligible accuracy change (0.3%), and outperforms random budgeting in both length and accuracy.

### 5.5. Generalization across different LMMs

We evaluate LMMs from different families and at different scales to verify VisionPulse's generalization. Table 3 demonstrates that VisionPulse generalizes across different LMM backbones under 5% retention. On both Qwen3-VL-Thinking 8B and InternVL-3.5-Thinking 4B, FastV consistently degrades accuracy with increasing generation length (e.g., Avg $A$ drops by 14.4% and 12.0%, while Avg $L$ increases by 19.3% and 11.0%). In contrast, FastV-VP preserves accuracy while reducing generation length: it slightly improves Avg $A$ (+0.2/+0.8%) with shorter Avg $L$ (-7.9/-12.4%), indicating stable gains across different LMMs.

*Table 3.* **Generalization across LMM backbones.** The results of VisionPulse on RealWorldQA and MIA-Bench for different LMMs. FastV (Chen et al., 2024b) is included for comparision.

| Method | RealWorldQA | | MIA-Bench | | Avg $L\downarrow$ | Avg $A\uparrow$ |
|---|---|---|---|---|---|---|
| | $L\downarrow$ | $A\uparrow$ | $L\downarrow$ | $A\uparrow$ | | |
| **Qwen3-VL-Thinking 8B** | | | | | | |
| **Baseline** | 408.4 | 77.24 | 1325.2 | 94.19 | 866.8 | 85.72 |
| **Retain $\leq$ 5% Tokens** | | | | | | |
| **FastV** | 348.1 | 57.52 | 1719.8 | 89.29 | 1033.95 | 73.41 |
| | -14.8% | -25.5% | +29.8% | -5.20% | +19.3% | -14.4% |
| **FastV-VP** | 400.0 | 76.60 | 1196.5 | 95.13 | 798.2 | 85.87 |
| | -2.1% | -0.8% | -9.7% | +1.0% | -7.9% | +0.2% |
| **InternVL-3.5-Thinking 4B** | | | | | | |
| **Baseline** | 3707.5 | 54.38 | 1756.9 | 89.70 | 2732.2 | 72.04 |
| **Retain $\leq$ 5% Tokens** | | | | | | |
| **FastV** | 3787.9 | 44.18 | 2274.5 | 82.63 | 3031.2 | 63.41 |
| | +2.2% | -18.8% | +29.5% | -7.9% | +11.0% | -12.0% |
| **FastV-VP** | 3246.5 | 55.16 | 1537.2 | 90.10 | 2391.9 | 72.63 |
| | -12.4% | +1.4% | -12.5% | +0.4% | -12.4% | +0.8% |

## 5.6. Effect of Different Importance Estimators

VisionPulse is a general framework for pruning during multimodal reasoning, emphasizing step-wise visual budget allocation rather than a specific token-importance estimator. To validate this, we integrate VisionPulse with FastV (Chen et al., 2024b) and Pdrop (Xing et al., 2024), and consider an ablation variant (*Ll*) that performs layer-wise dynamic pruning (instead of token-level selection) for a direct comparison. We evaluate all three variants on RealWorldQA and MIA-Bench under two retention settings.

As shown in Table 4, VisionPulse consistently improves the trade-off between efficiency and accuracy across different importance estimators. Under the 5% retention setting, FastV-VP achieves the best overall balance, reducing the average generation length by 19.6% while slightly improving accuracy, whereas Pdrop-VP achieves larger length reductions but incurs a modest accuracy drop. Notably, the layer-wise ablation Ll-VP achieves a comparable efficiency and accuracy trade-off as the token-level variant, suggesting that pushing the strategy to the layer level offers no clear advantage over token-level dynamic pruning. Overall, these results demonstrate that VisionPulse is compatible with different importance estimators, and that the primary gains stem from token-level dynamic budgeting.

## 5.7. Efficiency Analysis

Beyond reducing the generated output length, VisionPulse improves inference efficiency through maintaining dynamic visual sparsity during decoding. Specifically, we compare the original dense attention with our dynamic visual sparse attention, and report the end-to-end latency under different context lengths. As shown in Figure 5, VisionPulse consistently achieves lower latency yielding up to **1.30×** speedup

*Table 4.* **Effect of different importance estimators.** Results on RealWorldQA and MIA-Bench using different importance estimators within VisionPulse (VP).

| Method | RealWorldQA | | MIA-Bench | | Avg $L\downarrow$ | Avg $A\uparrow$ |
|---|---|---|---|---|---|---|
| | $L\downarrow$ | $A\uparrow$ | $L\downarrow$ | $A\uparrow$ | | |
| **Upper Bound (Retain 100% Tokens))** | | | | | | |
| **Baseline** | 678.0 | 72.81 | 2231.3 | 93.44 | 1454.7 | 83.12 |
| **Retain $\leq$ 10% Tokens** | | | | | | |
| FastV-VP | 578.0 | 73.20 | 1870.8 | 93.99 | 1224.4 | 83.60 |
| | -14.7% | +0.5% | -16.2% | +0.6% | -15.8% | +0.6% |
| Pdrop-VP | 586.3 | 72.02 | 2081.7 | 93.98 | 1334.0 | 83.00 |
| | -13.5% | -1.1% | -6.7% | +0.6% | -8.3% | -0.1% |
| Ll-VP | 545.2 | 73.20 | 2045.1 | 93.42 | 1295.2 | 83.31 |
| | -19.6% | +0.5% | -8.3% | -0.0% | -11.0% | +0.2% |
| **Retain $\leq$ 5% Tokens** | | | | | | |
| FastV-VP | 538.0 | 72.54 | 1800.1 | 95.09 | 1169.0 | 83.82 |
| | -20.6% | -0.4% | -19.3% | +1.8% | -19.6% | +0.8% |
| Pdrop-VP | 471.2 | 71.63 | 1837.1 | 91.72 | 1154.2 | 81.68 |
| | -30.5% | -1.6% | -17.7% | -1.8% | -20.6% | -1.7% |
| Ll-VP | 571.5 | 73.85 | 1979.6 | 93.43 | 1275.6 | 83.64 |
| | -15.7% | +1.4% | -11.3% | -0.0% | -12.3% | +0.6% |

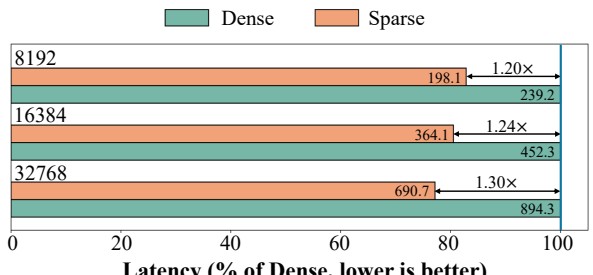

*Figure 5.* **End-to-end latency comparison of dense and sparse attention under different context lengths.** We set the batch size to 8 and the generation length to 1k tokens. Numbers on the bars indicate the actual latency (s).

over dense baseline. Overall, by explicitly aligning the visual budget with step-wise visual dependency, VisionPulse mitigates the coupled bottleneck in multimodal reasoning and delivers stable, practical end-to-end acceleration.

## 6. Conclusion

In this paper, we revisit visual token pruning and show that visual dependency during reasoning is inherently step-dependent. We further identify a coupled bottleneck where redundant visual context could steer LMM toward query-irrelevant regions, yielding longer and less reliable reasoning. To address these issues, we propose VisionPulse, a step-wise visual token pruning paradigm that estimates visual budget via a lightweight visual attention mass, and retains only critical visual tokens at each step. Experiments show that VisionPulse retains only 5% visual tokens per step with shortening reasoning traces, while keeping accuracy almost unchanged. These results establish VisionPulse as an extensible framework for test-time visual compression and for studying its interplay with multimodal CoT generation.

## Impact Statement

This paper presents work whose goal is to advance the field of Machine Learning by enabling improved analysis and acceleration of complex reasoning in multimodal large language models. There are many potential societal consequences of our work, none which we feel must be specifically highlighted here.

## Acknowledgements

This work is partially supported by National Natural Science Foundation of China (62376274, 62437002). Zhiwu Lu is the corresponding author.

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

# A. Additional Analysis

## A.1. Visual Attention Mass vs. Activated Token Coverage

Figure 6 extends Figure 1 with step-wise visualizations that link the visual attention mass $M_{vis}^t$ to the spatial coverage of activated visual tokens. When the reasoning step is primarily language-driven (e.g., *so*), $M_{vis}^t$ is low and activation is sparse, indicating that only a small fraction of visual tokens is needed. In contrast, during visually grounded steps (e.g., *square*), attention spreads over a broader set of tokens and higher-percentile activations cover substantially larger regions, accompanied by a clear increase in $M_{vis}^t$. These examples qualitatively support our finding that $M_{vis}^t$ correlates with the number of effectively activated tokens, motivating mass-guided dynamic budgeting.

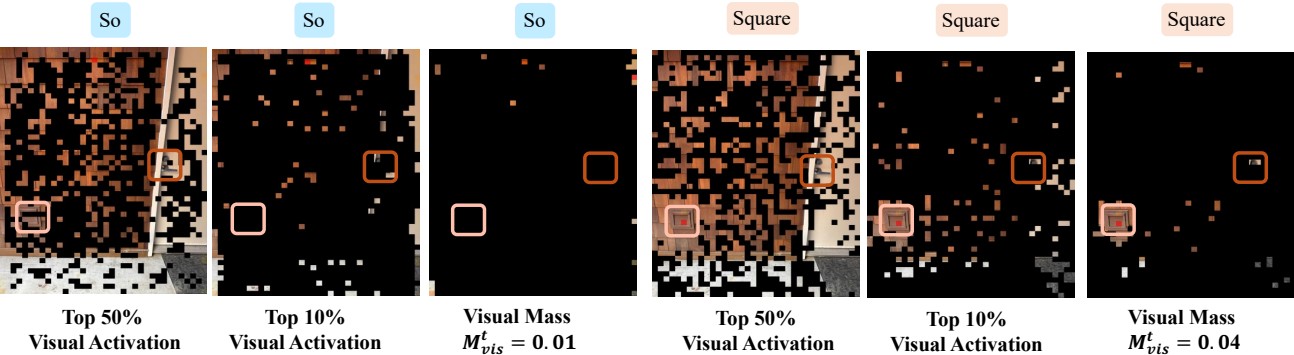

*Figure 6.* **Visual attention mass vs. activated token coverage.** We visualize step-wise visual activation at representative decoding steps. For each step, we show the regions covered by the top-50% and top-10% activated visual tokens, together with the corresponding visual attention mass $M_{vis}^t$. The examples illustrate that larger $M_{vis}^t$ is accompanied by broader token activation, while low-mass steps exhibit sparse, localized activation.

## A.2. Effects of different Budgeting strategies

We study how different budgeting strategies affect step-wise retention under a controlled setting, using Top-$p$ with $p = 0.10$, Top-$k$ with $k = 256$, and temperature scaling $\tau = 0.4$. As shown in Figure 7, Top-$k$ and Top-$p$ strategies are poorly matched to the step-wise variation of visual dependency. Top-$k$ allocates a constant budget across steps, and thus cannot increase retention when the model requires broader visual evidence, nor reduce retention when the step is largely language-driven. Interestingly, Top-$p$ exhibit a trend opposite to the visual-attention variation: since softmax produces dense distributions with an inherent long tail (Ren et al., 2020; Martins & Astudillo, 2016), even low-visual-dependency steps can accumulate non-trivial probability mass over many tokens, causing Top-$p$ to retain an unnecessarily large set and inflate the budget. In contrast, visual mass-guided budgeting directly tracks step-wise visual demand and yields a more stable retention behavior aligned with the visual reliance.

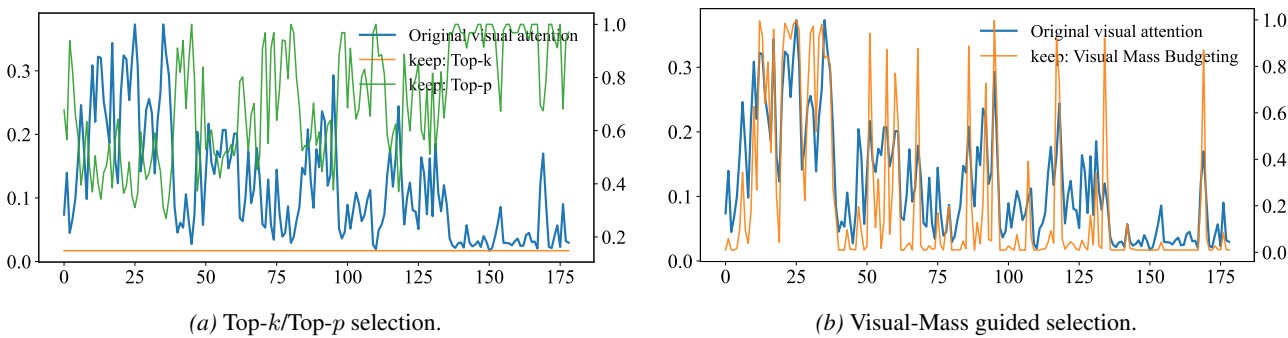

*(a)* Top-$k$/Top-$p$ selection.

*(b)* Visual-Mass guided selection.

*Figure 7.* **Budget and retention-ratio dynamics.** We compare the step-wise retention behavior under different budgeting strategies. The blue curve shows the original visual attention mass, while the colored curves indicate the retained token ratio determined by (a) fixed Top-$K$/Top-$p$ selection and (b) Visual-Mass budgeting. Visual-Mass budgeting tracks attention fluctuations more closely, allocating higher budgets at visually grounded steps and lower budgets at language-dominant steps.

*Table 5.* Complementarity between prefill-stage token reduction and VisionPulse. Results are averaged over three benchmarks. FastV is applied with a prefill compression ratio of 0.5, and VisionPulse further performs step-wise token sparsification during decoding.

| Method | Avg $L\downarrow$ | Avg $A\uparrow$ |
|---|---|---|
| FastV | 2094.7 | 69.4 |
| **FastV + VisionPulse** ($\leq 10\%$) | 1967.4 | 70.4 |
| **FastV + VisionPulse** ($\leq 5\%$) | 1957.0 | 69.8 |

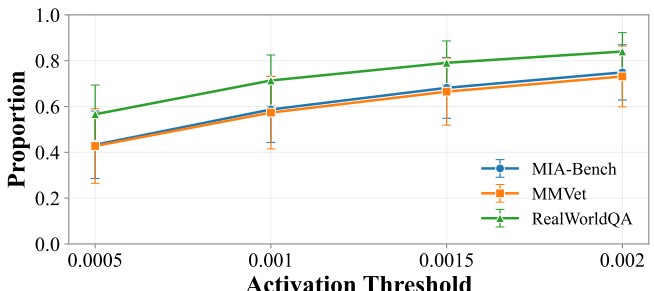

*Figure 8.* Proportion of never-activated visual tokens under different activation thresholds. Across MIA-Bench, MMVet, and RealWorldQA, many visual tokens remain inactive during decoding, indicating the existence of persistent visual redundancy during multimodal reasoning.

### A.3. Complementarity with Prefill Pruning

**Complementarity with prefill-stage token reduction.** VisionPulse performs step-wise dynamic pruning during decoding rather than at the prefill stage, it naturally raises the question of whether it can be effectively combined with existing pruning methods applied during prefill. Since prefill-stage pruning removes globally redundant visual tokens before decoding, it may offer additional acceleration when coupled with our decoding-time dynamic visual sparsification.

To examine this complementarity, we integrate FastV with VisionPulse and evaluate the combined method on MMVet, RealWorldQA, and MIA-Bench. Specifically, FastV is first applied during the prefill stage with a compression ratio of 0.5, after which VisionPulse performs step-wise visual token sparsification over the remaining visual tokens during decoding. The averaged results across the three datasets are reported in Table 5. FastV+VisionPulse consistently outperforms FastV alone, indicating that VisionPulse complements prefill-stage token reduction. This suggests that VisionPulse is orthogonal to static visual token pruning methods such as FastV and can be combined with them for additional efficiency improvements.

**Persistent visual redundancy during reasoning.** Beyond verifying the complementarity between VisionPulse and prefill-stage pruning, we further examine the temporal behavior of visual token usage during decoding. Specifically, we explore whether some visual tokens remain consistently inactive throughout the reasoning process. For each visual token, we measure its attention activation across all decoding steps and regard it as persistently inactive if its activation remains below a threshold $\delta$ at every step. As shown in Figure 8, across different datasets and threshold choices, a substantial fraction of visual tokens are rarely or never activated during reasoning. This observation suggests that multimodal reasoning contains a persistent inactive visual subset, which in principle can be removed without affecting the reasoning process. However, existing one-shot prefill-stage pruning methods are not specifically designed to exploit this temporal structure. These methods make the pruning decision before decoding begins, relying primarily on an initial static estimate of token importance to determine the retained visual subset. As a result, the retained subset may still include visual tokens that remain inactive in later reasoning steps, while some discarded tokens may become relevant as the decoding process evolves.

## B. Implementation Details

### B.1. Experimental Setup

We mainly evaluate VisionPulse on Qwen3-VL-Thinking-4B under aggressive visual-token retention. Unless otherwise stated, all results are obtained with decoding temperature 0.7 and a maximum generation length of 16,384 tokens. In Equation (6), we set the temperature scaling parameter $\tau$ to 0.4 for the 10% retention setting and to 0.1 for the 5% retention setting. For each benchmark, we follow the standard evaluation protocol and report accuracy (%) together with the average generation length $L$. All inference experiments are conducted on 8× NVIDIA A800 80GB GPUs.

### B.2. Datasets and Evaluation Protocols

We evaluate on seven multimodal reasoning benchmarks: CharXiv RQ (Wang et al., 2024), InfoVQA (Mathew et al., 2022), ChartQA (Masry et al., 2022), MMStar (Chen et al., 2024a), RealWorldQA (xAI, 2024), MMVet (Yu et al., 2024), and MIA-Bench (Qian et al., 2025). All evaluations are conducted with VlMEvalKit (Duan et al., 2024) using the default

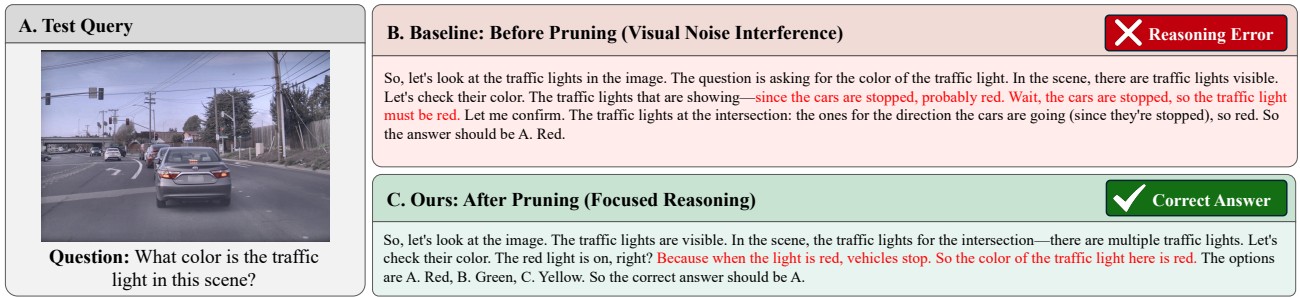

Figure 9. **Qualitative example of visual-noise interference on unnecessary reasoning.** Given the same query, the full-context baseline is distracted by query-irrelevant visual cues (e.g., stopped cars) and produces an incorrect reasoning trace, whereas VisionPulse (C) prunes redundant visual tokens during decoding, focuses attention on the traffic light, and yields the correct answer.

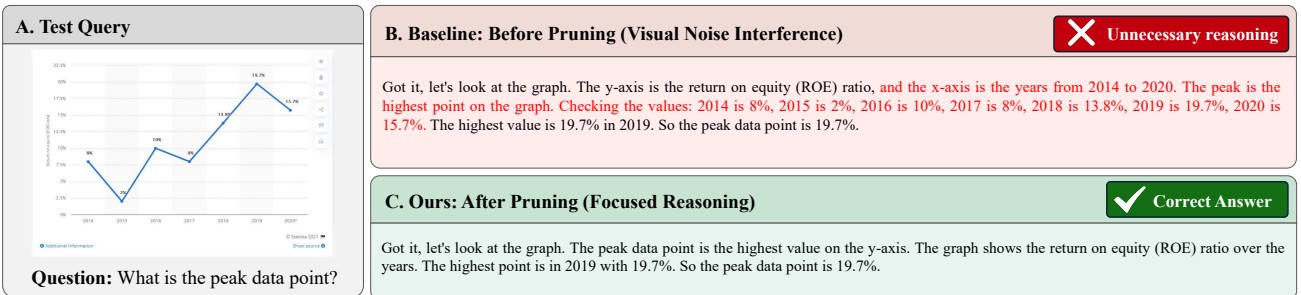

Figure 10. **Qualitative example of visual-noise interference on unnecessary reasoning.** On chart-based queries, the full-context baseline (B) tends to produce verbose, query-irrelevant descriptions by enumerating many values in the plot, resulting in unnecessary reasoning and longer generation. In contrast, VisionPulse (C) prunes redundant visual context during decoding, keeps attention on the peak point required by the question, and answers correctly with a shorter, more focused reasoning trace.

prompts and scoring rules. For LLM-judged benchmarks, we use GPT-4o for MIA-Bench and GPT-4o-mini for MMVet, CharXiv RQ, and MMStar.

**CharXiv RQ.** CharXiv (Wang et al., 2024) is a realistic chart-understanding benchmark curated from scientific papers, comprising 2,323 natural charts with associated questions. It emphasizes fine-grained visual grounding and domain-specific comprehension, covering both descriptive chart reading and reasoning-intensive queries. We use the reasoning validation subset as CharXiv RQ to evaluate the performance of our method, follow the default CharXiv evaluation in VlMEvalKit.

**InfoVQA.** InfoVQA (InfographicVQA) (Mathew et al., 2022) focuses on question answering over infographics that required methods to jointly reason over the document layout, textual content, graphical elements, and data visualizations. It contains 5,485 infographic images and 30,035 question–answer pairs, with diverse sources and strong layout/OCR dependence. InfoVQA is widely used to evaluate visual-dominant multimodal document understanding that requires integrating reading, layout cues, and visual symbols. We follow the default InfoVQA evaluation in VlMEvalKit.

**ChartQA.** ChartQA (Masry et al., 2022) tests question answering about charts with both visual and logical reasoning, often requiring numerical comparison and aggregation grounded in chart elements. It includes 9.6K human-authored questions and 23.1K machine-generated questions focusing on logical and visual reasoning questions, which involve complex reasoning and visual references to charts. We follow the default ChartQA evaluation in VlMEvalKit.

**MMStar.** MMStar (Chen et al., 2024a) is a vision-indispensable benchmark designed to reduce shortcut solving and ensure that visual evidence is necessary for answering. It comprises 1,500 human-curated challenge samples organized into 6 core capabilities and 18 fine-grained axes for diagnostic evaluation. Strict human review is involved to ensure each selected sample exhibits visual dependency, minimal data leakage, and requires advanced multi-modal capabilities for the solution. We follow the default MMStar evaluation in VlMEvalKit.

**RealWorldQA.** RealWorldQA (xAI, 2024) evaluates real-world visual understanding with an emphasis on basic spatial and physical reasoning. The initial release contains over 700 images, each paired with a question and an easily verifiable answer, including anonymized vehicle-captured scenes. It is commonly used to test grounding in everyday real-world contexts rather than memorized or purely language-driven behavior. We follow the default RealWorldQA evaluation in VlMEvalKit.

**A. Test Query**

**Question:** Are we ascending or descending?

**B. Baseline: Before Pruning (Visual Noise Interference)** ❌ **Unnecessary reasoning**

So, let's try to figure out if we're ascending or descending. The image shows a street with houses on both sides. On the right side, the houses are built on a slope, and the street seems to be going uphill. Wait, when you're driving, if the road is going up, you're ascending. But let's look at the perspective. The road in the image is a straight road, Wait, maybe the key is the direction but the houses on the right are higher up, and the left side has trees and a sidewalk of the road. If the road is sloping upward, then ascending. But how do we tell? Let's think about the view. If you're looking at the road, and the road is going up, then you're ascending. In the image, the road seems to be going up because the houses on the right are higher, and the road is inclined. Wait, but maybe it's the other way. Wait, no—if you're driving, and the road is going up, you're ascending. Let's check the options. The question is "Are we ascending or descending?" So we need to determine the direction of the road. Looking at the image, the road is a steep incline. Wait, actually, in the image, the road appears to be going uphill. Wait, but maybe not. Wait, let's think about the perspective. If the road is going up, then the cars in front would be going up. But in the image, there's a red car ahead. Wait, maybe the key is the slope. The right side has a hedge that's higher, so the road is going up. So if we're on the road, and the road is going up, then we're ascending. So the answer would be A. Ascending.

**C. Ours: After Pruning (Focused Reasoning)** ✅ **Correct Answer**

So, let's look at the image. The road is a hill, and we need to figure out if we're going up or down. In the image, the perspective shows that the road is sloping upwards as we look up towards the distance. Wait, but when you're driving, if the road is going up, you're ascending. Let me check the options. The question is "Are we ascending or descending?" So, if the road is going up, then ascending? Let's see the image: the road seems to be going uphill. So the answer should be A. Ascending.

*Figure 11.* **Qualitative example of redundant visual reasoning.** In this example, the LMM identifies the correct reasoning direction early on, but is later distracted by query-irrelevant visual cues (e.g., buildings and trees), which triggers repetitive reasoning. As a result, it follows a prolonged trajectory before eventually returning to the same conclusion implied by its initial reasoning.

**MMVet.** MMVet (Yu et al., 2024) is an open-ended benchmark designed to evaluate integrated vision-language skills on complicated multimodal tasks, such as recognition, OCR, spatial reasoning, knowledge, and arithmetic It contains 200 images and 218 questions, each annotated with the capabilities required for solving. We follow the default MMVet evaluation in VlMEvalKit.

**MIA-Bench.** MIA-Bench (Qian et al., 2025) focuses on complex multimodal instruction following, emphasizing strict adherence to layered, constraint-heavy prompts in open-ended generation. It comprises 400 image–prompt pairs crafted to elicit instruction violations (e.g., formatting constraints, multi-part requirements, and compositional conditions). MIA-Bench is commonly used to evaluate controllability and instruction fidelity of MLLMs under complex prompts. We follow the default MIA-Bench evaluation in VlMEvalKit.

## C. More Qualitative Analysis for The Coupled Bottleneck in LMMs

In this section, we provide additional qualitative examples (Figures 9 to 11) to analyze how query-irrelevant visual evidence induces unnecessary reasoning and contributes to the coupled bottleneck in LMMs.

## D. More Qualitative Analysis of Different Pruning Paradigms

To provide a more intuitive qualitative analysis of different pruning paradigms, we visualize both the reasoning trajectories and the retained visual tokens under 5% visual token retention. As shown in Figure 12, static pruning methods struggle to preserve task-relevant visual information, making them inadequate for multimodal reasoning and often leading to incorrect predictions. In contrast, VisionPulse dynamically retains the visual tokens required at each reasoning step, thereby preserving critical information and enabling correct reasoning. This observation is also consistent with our discussion that, during the prefill stage, the model tends to prioritize generating the first token rather than acquiring all the visual evidence required for subsequent reasoning.

**Test Query**

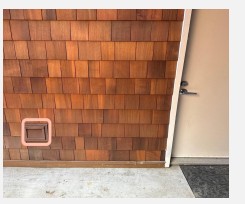

**Question:** Where is the brown square vent relative to the door?

**Retained Visual Activation**

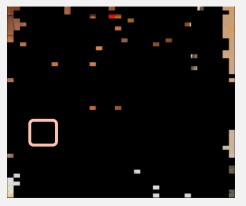

**Retained visual tokens**

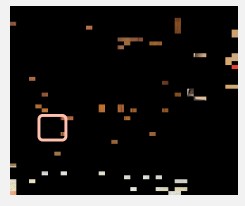

**Baseline**

So, let's look at the image. The door is on the right side of the image, with the beige color and the handle. The brown square vent is on the wooden shingle wall. Let's check the position. The vent is on the left side of the door, because the door is on the right, and the vent is part of the wall next to the door but to the left of it. Wait, the door is the beige part with the handle. The vent is on the wooden wall, which is adjacent to the door. So the vent is to the left of the door. Let's see the options. Option B says the brown square vent is to the left of the door. That seems correct. Option A says there's no vent, but there is a brown square vent. So A is wrong. C says right, but the vent is left. So B is correct.

**VisionPulse**

So, let's look at the image. The door is on the right side of the picture. The brown square vent is on the wall to the left of the door. Let's check the options. Option B says the vent is to the left of the door. That matches. Option A says there's no vent, but there is one. Option C says right, which is wrong. So the answer is B.

**VisionZip**

So, let's look at the image. The question is about the brown square vent relative to the door. First, check if there's a brown square vent. The image shows a door on the right side, and the wall is covered with wooden shingles (brown). Wait, the options mention a "brown square vent"—but in the image, is there a vent? Let me check again. The image has a door with a doorknob, and the wall is shingles. There's no visible vent in the image. So option A says "There is no brown square vent." That must be correct. Let me confirm: the options are A, B, C. The image doesn't show a vent, so A is the answer.

**FastV**

So, let's look at the image. The question is about the brown square vent's position relative to the door. First, I need to check if there's a brown square vent. The image shows a wall with a door on the right side. Wait, the door is on the right, and the vent—wait, is there a vent? Let me check again. The image has a door, and maybe a vent? Wait, the options say A is no vent. Let me see. The image: the door is on the right, and the left side of the image is the wall. Wait, maybe there's a vent on the wall. Wait, looking at the image, the door is on the right, and the wall to the left of the door—wait, no, the door is part of the wall. Wait, maybe the brown square vent is on the wall. Wait, the options: A says no vent. Let me check if there's a vent. The image shows a door, and the wall. Is there a square vent? Let's see. The image has a door, and the area around it. Wait, maybe the vent is not present. Wait, the options: A is no vent. Let me check again. The image: the door is on the right, and the left side of the image is the wall. Wait, maybe there's a vent. Wait, no, maybe there's no brown square vent. Wait, the image shows a door, and the wall. Let me look for a square vent. Hmm. Wait, maybe the vent is not visible. Wait, the options: A says there is no brown square vent. Let's confirm. The image: the door is on the right, and the wall. Is there a square vent? Let's see. The image has a door, and the wall. Maybe there's no vent. So option A would be correct.

*Figure 12.* **Visualization comparison of the reasoning trajectories of different visual token pruning methods.** We further visualize the retained visual tokens of VisionZip (dominant tokens) and FastV under 5% visual token retention.

