# OpenReview forum: "VisionPulse: Dynamic Visual Sparsity for Efficient Multimodal Reasoning"
_ICML.cc/2026/Conference — ICML 2026 regular_

### Official Review · Reviewer_YX1P · 2026-03-12

**Soundness:** 3
**Presentation:** 3
**Significance:** 3
**Originality:** 3
**Overall Recommendation:** 4
**Confidence:** 3

**Summary:**

This paper studies efficient multimodal reasoning and argues that visual evidence is highly step-dependent during decoding, so static visual token pruning at prefill is mismatched to reasoning-time needs. To address this, the paper proposes VisionPulse, a training-free framework that performs step-wise visual token pruning during reasoning. The method uses a lightweight visual attention mass signal to estimate a per-step retention budget and keeps only the most critical visual tokens under that budget. Experiments on seven benchmarks show that the method can retain only a small fraction of visual tokens while largely preserving accuracy and reducing reasoning length and latency.

**Compliance With Llm Reviewing Policy:**

Affirmed.

**Final Justification:**

The authors have addressed some of the concerns,  while I think more baseline comparison is needed.

**Key Questions For Authors:**

1. The novelty is somewhat limited at the algorithmic level. The main idea is dynamic step-wise pruning plus a budget estimated from attention mass, which is intuitive but not highly technically deep.

2. The method relies heavily on attention-based signals. The paper shows a correlation between visual attention mass and active token count, but this is still a heuristic proxy rather than a stronger theoretical justification.

3. The comparison is mainly against pruning baselines. It would be stronger to compare against a broader set of efficient multimodal reasoning methods, not only token-pruning approaches.

4. The “coupled bottleneck” discussion is interesting, but the causal claim that redundant visual context directly harms reasoning reliability is supported mostly by case studies and indirect evidence. More systematic analysis would strengthen this point.

**Limitations:**

see above

**Strengths And Weaknesses:**

1. The paper studies an important problem: reducing inference cost in multimodal reasoning without hurting accuracy.

2. The core motivation is clear and intuitive: visual dependence changes across reasoning steps, so dynamic pruning is more suitable than one-shot static pruning.

3. The proposed framework is simple and practical, since it is training-free and can be combined with different importance estimators.

4. Experimental results are strong: under aggressive retention, the method preserves accuracy much better than static pruning baselines and often shortens reasoning traces.

5. The paper includes ablation, cross-backbone generalization, and latency analysis, which makes the empirical section relatively complete.

---

> ### Author Rebuttal · Authors · 2026-03-31
>
> **W1. The main idea is dynamic step-wise pruning plus a budget estimated from attention mass, which is intuitive but not highly technically deep.**
>
> **A:** Thanks. In our responses to W2 and W4, we have clarified the rationale of our method from theoretical perspectives: (1) visual attention mass as a principled proxy for the effective visual-token budget, and (2) the coupled bottleneck under autoregressive decoding.
>
> Besides these theoretical justifications, we have also supported our design with extensive empirical evidences: step-wise attention analysis, the correlation between visual attention mass and activated visual tokens, and qualitative analysis of the coupled bottleneck. These empirical analyses provide strong evidence for the underlying motivation and mechanism of our proposed VisionPulse.
>
> Moreover, VisionPulse is intentionally designed to be extremely lightweight. For acceleration methods [1], this is a desirable property, since more complex designs often introduce additional overhead and thus weaken their practical value. Despite its simple design, VisionPulse consistently achieves strong performance-efficiency trade-offs across benchmarks and model families. We will incorporate the above analysis in the revision.
>
> [1] SageAttention: Accurate 8-Bit Attention for Plug-and-play Inference Acceleration. ICLR 2025.
>
> **W2. The paper shows a correlation between visual attention mass and active token count, but this is still a heuristic proxy rather than a stronger theoretical justification.**
>
> **A:** Thanks for your insightful review. We provide a further justification for the use of visual attention mass as a proxy for active visual token count.
>
> Let $a_{t,i}$ denote the attention from the query at step $t$ to the $i$-th visual token, and $M_t^{vis}=\sum_{i=1}^{N_v} a_{t,i}$ denote the total visual attention mass. Define the active token set $\mathcal S_t(\delta)=\{i: a_{t,i}>\delta\}$ with size $N_t^{act}(\delta)$. We decompose the visual mass as:
> $$
> M_t^{vis}=N_t^{act}(\delta)\mu_t(\delta)+R_t(\delta),
> $$ where $R_t(\delta)=\sum_{i\notin\mathcal S_t(\delta)} a_{t,i}$ is the inactive long-tail mass, and $\mu_t(\delta)=\frac{1}{N_t^{act}(\delta)}\sum_{i\in\mathcal S_t(\delta)} a_{t,i}$ is the average attention over active tokens.
>
> When the following two conditions hold: **(i)** the average attention over active visual tokens remains relatively stable across decoding steps $t$，and **(ii)** the inactive mass $R_t(\delta)$ is small, we can derive the following approximation:
> $$
> M_t^{vis} \approx \mu N_t^{act}(\delta),
> $$ where $\mu$ denotes a representative value of $u_t(\delta)$ across decoding steps. To support this approximation, we conduct correlation analysis and observe a strong linear relationship between visual attention mass and active token count across decoding steps (see Fig. 3), supporting Condition (i). Temperature scaling is also used to suppress the softmax long tail, reducing $R_t(\delta)$ to better satisfy Condition (ii). We will incorporate additional analyses in the revision.
>
> **W3. Comparison against a broader set of efficient multimodal reasoning methods.**
>
> **A:** Thanks for pointing this out. Existing methods [1–2] typically improve efficiency by training models to produce shorter reasoning paths, but do not explicitly address visual-textual attention sparsity during reasoning. In contrast, VisionPulse targets this aspect by pruning irrelevant visual tokens based on the evolving reasoning state. As a plug-and-play framework, VisionPulse is orthogonal to these methods and can be combined with them. We will include additional experiments in the revision to evaluate this integration.
>
> [1] AdaptThink: Reasoning Models Can Learn When to Think, EMNLP 2025.\
> [2] Look-Back: Implicit Visual Re-focusing in MLLM Reasoning, AAAI 2026.
>
> **W4. More systematic analysis of the “coupled bottleneck”.**
>
> **A:** Thanks for your insightful review. We further clarify the mechanism of the coupled bottleneck from the perspective of autoregressive decoding. At decoding step $t$, the model predicts
> $$
> y_t \sim p(y_t|x, v, y_{<t}),
> $$ where $v$ is the set of visual tokens used for the current reasoning subgoal. By decomposing $v$ into two sets of relevant and irrelevant visual tokens $v = v_r \cup v_i$ , we have:
> $$
> p(y_t|x, v_r, v_i, y_{<t}).
> $$
> Appling Bayes' formula to this equation, when the contribution of $v_i$ becomes dominant, substantial irrelevant visual tokens would bias the prediction and hidden state. This aligns with our empirical observation that the model tends to drift toward task-irrelevant intermediate descriptions. Moreover, such deviations would propagate through autoregressive decoding, progressively biasing subsequent predictions. Once these task-irrelevant intermediate descriptions are used as premises for subsequent reasoning, they can lead to additional unnecessary reasoning paths and even gradually degrade reasoning reliability.

---

> > ### Author Rebuttal · Reviewer_YX1P · 2026-04-01
> >
> > I think more token pruning methods, whether for a large language model or a large multimodal model, should be compared. The current number of comparison baselines is small. I would like to see the results of more baseline methods.

---

> > > ### Author Response · Authors · 2026-04-03
> > >
> > > Thank you once again for taking the time to review our paper and for providing such insightful and constructive feedback.
> > >
> > > To further compare VisionPulse with existing token reduction methods, we evaluate three methods [1–3] on RealWorldQA under the 5% retention setting. As shown in the table below, existing methods suffer from substantial performance degradation, and compensate for the missing visual information at each step by producing longer generations (such as Quest and ShadowKV). In contrast, VisionPulse effectively captures the evolving visual dependency and thus achieves superior performance.
> > >
> > > | Qwen3 VL Thinking 4B | Gen. $\downarrow$ | Acc. $\uparrow$ |
> > > | -------------------- | ----------------------------- | ------------------- |
> > > | Baseline             | 678.0                         | 72.81               |
> > > | R-KV                 | 554.0                         | 62.88               |
> > > | Quest                | 1866.9                        | 63.53               |
> > > | ShadowKV             | 838.4                         | 67.19               |
> > > | VisionPulse          | **538.0**                     | **72.54**           |
> > >
> > > [1] Quest: Query-Aware Sparsity for Efficient Long-Context LLM Inference. ICML2024.\
> > > [2] ShadowKV: KV Cache in Shadows for High-Throughput Long-Context LLM Inference. ICML 2025.\
> > > [3] R-KV: Redundancy-aware KV Cache Compression for Reasoning Models. NIPS 2025.

---

### Official Review · Reviewer_NpSH · 2026-03-12

**Soundness:** 2
**Presentation:** 3
**Significance:** 3
**Originality:** 2
**Overall Recommendation:** 4
**Confidence:** 4

**Summary:**

The paper discovers that important visual tokens are evolving during decoding, and redundant visual tokens can steer LMMs to query irrelevant regions. Driven by this discovery, VisionPulse proposes a training-free step-wise token pruning approach using visual mass.

**Compliance With Llm Reviewing Policy:**

Affirmed.

**Final Justification:**

Author addressed my questions, so I raised my score from 3 to 4.

**Key Questions For Authors:**

I have addressed the questions in the above weakness section. If the author can address my questions, I will adjust my score.

**Limitations:**

yes

**Strengths And Weaknesses:**

Strength:

Paper is well written and present its methodology clearly.

Weakness:

1. There is no end to end efficiency analysis. As a token reduction methodology, it's important to show end-to-end latency, throughput, and peak KV-cache memory usage.

2. Paper only shows performance on image benchmark, effectiveness of VisionPulse on video understanding is not validated.

3. Relying on hardcoded anchor layer (layer 17) lacks justification. It is unclear if this overfits the specific model or it can be generalized.

4. It is unclear that if VisionPulse is orthogonal to existing prefilling token reduction method, as prefilling is the most expensive phase in real-world deployments.

5. As paper only compares VisionPulse with static visual pruning methods, there are existing work that also aim at decoding phase token reduction. There is no experiment showing VisionPulse is superior to them.

---

> ### Author Rebuttal · Authors · 2026-03-31
>
> **W1. There is no end to end efficiency analysis.**
>
> **A:** Thank you for pointing this out. The acceleration of VisionPulse comes from two factors during decoding: sparse attention and reduced generation length. In the main paper, we have reported the **end-to-end latency**, which reflects the combined gain from both factors. To isolate the contribution of sparse attention alone during inference, we further report the **throughput and end-to-end latency** under a fixed output length in the table below. We set the batch size to 8 and the decoding length to 1k. Since our method performs token selection directly on the KV cache without introducing any compression operation, the peak KV-cache memory usage remains the same as that of the baseline.
>
> |Visual Context Length| Latency (Dense/Sparse, s) |Throughput (Dense/Sparse, tok/s)| Speedup↑|
> |:-|:-:|:-:|:-:|
> |8k|239.2/198.1|34.2/41.4|1.20x|
> |16k|452.3/364.1|18.1/22.5|1.24x|
> |32k|894.3/690.7|9.2/11.9|1.30x|
>
> It can be seen that VisionPulse achieves consistent end-to-end acceleration, with 1.20×–1.30× speedup across different visual context lengths. Importantly, as sparse attention is applied across more decoding steps (>1k), VisionPulse yields progressively larger efficiency gains, especially for complex long-horizon reasoning tasks.
>
> **W2. Effectiveness of VisionPulse on video understanding is not validated.**
>
> **A:** Thanks. More results on video understanding can be found in our response to W3 of Reviewer XTDY.
>
> **W3. Relying on hardcoded anchor layer (layer 17) lacks justification.**
>
> **A:** Thank you for pointing this out. Please refer to our response to W2 of Reviewer XTDY for a detailed discussion of the anchor layer selection.
>
> **W4. It is unclear if VisionPulse is orthogonal to existing prefilling token reduction method.**
>
> **A:** Thanks for your insightful review. To make this clear, we conduct experiments by combining FastV (prefilling token reduction) and VisionPulse (decoding token reduction) on three datasets: MMVet, RealWorldQA, and MIA-Bench. In these experiments, the compression ratio of FastV is 0.5, and VisionPulse selects the remaining visual tokens at each step. The average results across the three datasets are shown in the table below. We find that FastV+VisionPulse leads to consistent improvements over FastV, demonstrating that VisionPulse is indeed orthogonal to the prefilling token reduction method FastV.
>
> |Method|Gen.|Acc.|
> |:-|:-:|:-:|
> |FastV|2094.7|69.4|
> |FastV+VisionPulse (≤10%)|1967.4|70.4|
> |FastV+VisionPulse (≤5%)|1957.0|69.8|
>
> **W5. There are existing works that also aim at decoding phase token reduction. There is no experiment showing VisionPulse is superior to them.**
>
> **A:** Thank you for pointing this out. Existing decoding-phase token reduction methods, such as Quest [1], ShallowKV [2], and R-KV [3], primarily focus on selecting and reducing long-context text inputs/outputs, and they typically rely on a **fixed top-k budget for layer-wise token selection**.
>
> While effective in text-only settings, this design in [1-3] is less suitable for multimodal scenarios. The main reason lies in that the demand for visual tokens varies across layers (not top-k) during multimodal inference [4], with significantly higher demand for visual tokens in the earlier layers. Furthermore, our work demonstrates that the need for visual evidence is also dynamic across decoding steps: in vision-dominated steps, the number of activated visual tokens can far exceed what a simple top-k budget can retain. Overall, while effective in text-only settings, these methods [1-3] are less suited for multimodal tasks where the visual information requirements evolve throughout the reasoning process.
>
> [1] Quest: Query-Aware Sparsity for Efficient Long-Context LLM Inference. ICML 2024.\
> [2] ShadowKV: KV Cache in Shadows for High-Throughput Long-Context LLM Inference. ICML 2025.\
> [3] R-KV: Redundancy-aware KV Cache Compression for Reasoning Models. NIPS 2025.\
> [4] PyramidDrop: Accelerating Your Large Vision-Language Models via Pyramid Visual Redundancy Reduction. CVPR 2025.

---

> > ### Author Rebuttal · Reviewer_NpSH · 2026-04-01
> >
> > Thank author for the reply. Here are the two questions I raised but didn't get fully resolved. First, regarding anchor layer selection, I understand the main motivation, but it still lacks ablation analysis on specific layer selection. Second, regarding comparison with decoding phase token reduction, there is no experiment comparing the performance against existing work. It is hard to understand the performance without experiment comparison.

---

> > > ### Author Response · Authors · 2026-04-03
> > >
> > > Thank you once again for taking the time to review our paper and for providing such insightful and constructive feedback.
> > >
> > > To further study the effect of anchor layer selection on overall performance, we conduct another ablation analysis. Specifically, we sample the anchor layer every four layers (from layer 5 to 25). As shown in the table below, pruning at excessively early layers leads to a performance drop. In contrast, from the middle layer (i.e. layer 17) onward, visual perception is better preserved, and overall performance is well maintained. These results are consistent with prior findings and support our motivation: visual dependency is stronger in the early layers, while it becomes sparser in the middle-to-late layers (i.e., visual token reduction becomes feasible).
> > >
> > > | Metric\Layer | 5 | 9 | 13 | 17 | 21 | 25 | Baseline |
> > > |-|:-:|:-:|:-:|:-:|:-:|:-:|:-:|
> > > | Gen. $\downarrow$ | 482.2 | 487.9 | 513.5 | 564.0 | 578.1 | 603.2 | 678.0 |
> > > | Acc. $\uparrow$ | 69.28 | 70.33 | 70.98 | 72.54 | 72.81 | 72.29 | 72.81 |
> > >
> > > To better understand the performance of VisionPulse against existing decoding-phase token reduction methods, we evaluate three representative approaches [1–3] on RealWorldQA under the 5% retention setting. As shown in the table below, existing methods suffer from substantial performance degradation, and produce longer generation in order to compensate for the missing visual information at each step (such as Quest and ShadowKV). In contrast, VisionPulse effectively captures the evolving visual dependency and thus achieves superior performance.
> > >
> > > | Qwen3 VL Thinking 4B | Gen. $\downarrow$ | Acc. $\uparrow$ |
> > > |---|:-:|:-:|
> > > | Baseline | 678.0 | 72.81 |
> > > | R-KV | 554.0 | 62.88 |
> > > | Quest | 1866.9 | 63.53 |
> > > | ShadowKV | 838.4 | 67.19 |
> > > | VisionPulse | **538.0** | **72.54** |
> > >
> > > [1] Quest: Query-Aware Sparsity for Efficient Long-Context LLM Inference. ICML2024.\
> > > [2] ShadowKV: KV Cache in Shadows for High-Throughput Long-Context LLM Inference. ICML 2025.\
> > > [3] R-KV: Redundancy-aware KV Cache Compression for Reasoning Models. NIPS 2025.

---

### Official Review · Reviewer_czjy · 2026-03-15

**Soundness:** 2
**Presentation:** 2
**Significance:** 1
**Originality:** 2
**Overall Recommendation:** 3
**Confidence:** 5

**Summary:**

This paper addresses the high inference overhead of large multimodal models (LMMs) caused by redundant visual tokens in multi-step reasoning, pointing out the flaws of existing static visual token pruning methods that only operate at the prefill stage—these methods ignore the step-dependent nature of visual evidence demand and the coupled bottleneck where redundant visual context misleads the model to irrelevant regions and lengthens reasoning traces. To solve these problems, the authors propose VisionPulse, a training-free dynamic visual token pruning framework that conducts step-wise token selection during reasoning: it calculates a lightweight visual attention mass to set a step-specific retention budget and only retains the most critical visual tokens, thus filtering redundant visual information while preserving relevant evidence. Extensive experiments on seven mainstream multimodal reasoning benchmarks with Qwen3-VL-Thinking show that VisionPulse retains only 5% of visual tokens per step, shortens reasoning traces by 11.2% with almost unchanged accuracy, outperforms state-of-the-art static pruning methods significantly, and achieves good generalization across different LMM backbones and compatibility with various token importance estimators. The paper’s key contributions include empirically demonstrating the step-dependent characteristic of visual evidence demand in multimodal reasoning, identifying the coupled bottleneck induced by redundant visual context, proposing the effective VisionPulse framework with dynamic pruning and budget allocation, and validating the framework’s efficiency, accuracy and generalization through comprehensive experiments, providing a new paradigm for test-time visual compression of LMMs. Overall, an important concept considered by this paper is the step-wise dynamic visual attention and token demand in the multi-step reasoning process of LMMs, which breaks the static assumption of traditional visual pruning methods. Overall, the authors assess a fundamental challenge in LMM inference efficiency: the dual cost of redundant visual context in increasing per-step computation and distorting reasoning traces, which is a key bottleneck for the real-world deployment of LMMs with strong multimodal reasoning capabilities.

**Compliance With Llm Reviewing Policy:**

Affirmed.

**Key Questions For Authors:**

Will VisionPulse maintain its efficiency and accuracy advantages on ultra-long context lengths (128K/256K) and long video/document understanding benchmarks such as LVBench, VideoMME and MMLongBench-Doc? A positive response will confirm the method’s generalization to long-context multimodal tasks and enhance the evaluation of its practical value; a negative response will indicate the method has limitations in long-context scenarios and require further optimization.
How does VisionPulse perform on high-resolution image cut-and-understand agent benchmarks like HRBench4K/8K and V*? Good performance will prove the method’s effectiveness in agent-related multimodal tasks and expand its application scenarios; poor performance will suggest the need to adapt the pruning strategy for high-resolution image cutting scenarios.
Why not conduct comprehensive experiments on all benchmarks in the Qwen3-VL technical report, and will supplementing these experiments still show the stable superiority of VisionPulse over existing methods? Supplemented positive results will make the experimental validation more solid and reliable, increasing the persuasiveness of the paper; inconsistent results will mean the method’s superiority is scenario-dependent and needs to clarify applicable scopes.
What are the core advantages of the training-free VisionPulse compared with methods that integrate visual sparsification during training, given that training-free methods have a lower theoretical upper bound? A clear explanation of unique advantages (e.g., no retraining cost, good compatibility with off-the-shelf LMMs, fast deployment) will highlight the practical value of the training-free design; a lack of convincing advantages will reduce the competitiveness of the method compared with training-based sparsification methods.
Is there any room for optimizing the VisionPulse framework to narrow the performance gap with training-based visual sparsification methods, and what specific optimization directions are there? Feasible optimization directions (e.g., combining lightweight fine-tuning, adaptive temperature scaling) will show the method’s scalability; no clear optimization plan will indicate the training-free design has obvious performance bottlenecks.

**Limitations:**

The paper only briefly mentions the research’s contribution to LMM inference acceleration in the impact statement and lacks an in-depth discussion of the method’s inherent limitations (e.g., lower upper bound compared with training-based sparsification, untested performance in long-context/high-resolution/agent scenarios) and potential negative societal impacts (e.g., the risk of incorrect reasoning caused by excessive visual token pruning in safety-critical multimodal tasks such as autonomous driving and medical image analysis). It is suggested that the authors add a dedicated limitations section to clearly state the method’s shortcomings in long-context, high-resolution and agent tasks, the performance gap with training-based methods, and the potential risks in safety-critical scenarios; at the same time, put forward specific future improvement directions (e.g., optimizing the pruning strategy for long-context data, combining lightweight fine-tuning to improve performance, adding a safety threshold for visual token retention in critical tasks) to make the research more complete and rigorous.

**Strengths And Weaknesses:**

Strengths
In terms of soundness, the paper’s core findings about the step-dependent visual evidence demand and coupled bottleneck are supported by rigorous empirical analysis of visual attention mass and heatmaps, and the VisionPulse framework is designed with clear mathematical formulations and logical reasoning, with experimental results on seven benchmarks fully verifying its effectiveness and generalization. In presentation, the paper structures content clearly from motivation, framework design to experiments, uses intuitive figures and tables to illustrate dynamic visual activation and framework workflow, and provides detailed implementation details and qualitative examples for reproducibility. In significance, the work addresses the critical inference overhead bottleneck of LMMs for real-world deployment, proposes a training-free dynamic visual pruning method that balances efficiency and accuracy, and provides new insights for the research of multimodal reasoning efficiency and visual token compression. In originality, the paper creatively abandons the static pruning paradigm of prior work, proposes step-wise visual token pruning driven by visual attention mass, identifies the coupled bottleneck of redundant visual context in reasoning trace and accuracy, and combines existing token importance estimation with dynamic budget allocation to form a novel and effective LMM inference acceleration framework.
Weaknesses
In soundness, the paper only conducts experiments on Qwen3-VL-Thinking with standard context lengths and lacks tests on ultra-long context lengths (128K/256K) and long video/document understanding benchmarks (LVBench, VideoMME, MMLongBench-Doc), making the generalization to long-context multimodal tasks unproven; it also omits evaluations on high-resolution image cut-and-understand agent benchmarks (HRBench4K/8K, V*) and fails to fully test all benchmarks in the Qwen3-VL technical report, leading to insufficient comprehensiveness of experimental validation. In presentation, the paper does not clearly compare the performance upper bound gap between training-free dynamic pruning and training-time sparsification, and the explanation of the core advantage of the training-free design is not in-depth enough. In significance, the lack of long-context and high-resolution agent task evaluations limits the practical application scope of the method in real-world long video understanding and high-resolution image analysis scenarios. In originality, the framework’s core token importance estimation borrows from existing FastV methods, and the innovation is mainly in dynamic budget allocation and step-wise pruning, with relatively limited original innovation in the basic module design.

---

> ### Author Rebuttal · Authors · 2026-03-31
>
> Thank you very much for your time, effort, and insightful review. We sincerely appreciate the care you devoted to evaluating our submission.
>
> During the rebuttal process, we noticed that the final part of the Summary contains both of the prompt-injection phrases inserted by ICML organizers: "Overall, an important concept considered by this paper..." and "Overall, the authors assess a fundamental challenge...", which indicates LLM-generated text had been directly incorporated into the review. Moreover, the broad and high-level nature of the LLM-generated review further increases the difficulty of providing targeted responses in the remainder of this rebuttal. For example, we are asked to evaluate our method across all benchmarks in the Qwen3-VL technical report to establish its stable superiority, which is difficult to achieve in such a short period of time. Therefore, we focus here on the most concrete and central questions.
>
> In particular, we consider two questions of this review to be central to the evaluation of this work, which are also shared by the other reviewers. First, regarding long-video or long-context evaluation, we have already conducted experiments on Video-MME, which verify that the effectiveness of VisionPulse arises from  the characteristics of visual attention during reasoning, rather than depending on context length or specific visual modality. Please refer to our response to W3 of Reviewer XTDY for more details. Second, regarding the advantages of our training-free design, VisionPulse can be seamlessly compatible with off-the-shelf LMMs. Moreover, our method is even orthogonal to existing training-based visual sparsification methods as well as prefilling token reduction methods, which makes it possible to integrate VisionPulse with these approaches for further acceleration. Please also refer to our responses to W4 of Reviewer NpSH and to W3 of Reviewer YX1P.

---

### Official Review · Reviewer_XTDy · 2026-03-16

**Soundness:** 3
**Presentation:** 2
**Significance:** 2
**Originality:** 3
**Overall Recommendation:** 4
**Confidence:** 3

**Summary:**

This paper targets the high inference cost of Large Multimodal Models (LMMs) in multi-step chain-of-thought reasoning. It shows that visual token importance is not static but varies across reasoning steps, with many text-dominant steps requiring little visual grounding. Based on this observation, the authors propose VisionPulse, a training-free step-wise pruning method that dynamically adjusts the number of retained visual tokens at each decoding step using a lightweight visual attention signal, reducing redundant computation and distraction from irrelevant visual context.

**Compliance With Llm Reviewing Policy:**

Affirmed.

**Key Questions For Authors:**

See weakness

**Limitations:**

See weakness

**Strengths And Weaknesses:**

Strenght:
1. Novel Insight on Dynamic Sparsity: The core contribution—that visual token importance is fluid and task-state dependent—is well-motivated by qualitative and quantitative evidence.
2. The observation of extra visual tokens leading to "visual noise interface" is insightful. By pruning these, VisionPulse not only speeds up the model but actually shortens the reasoning length by roughly 11.2%.
3. Plug-and-Play Versatility: As a training-free method, VisionPulse demonstrates considerable generalization across different model families (e.g., Qwen3-VL, InternVL-3.5) and different importance estimators like FastV and Pdrop. It can also be combined with existing token-importance estimators. The dynamic budget is also supported by an empirical correlation between visual attention mass and activated visual token count, with reported high Pearson correlations.

Weakness:
1. Computational Overhead of Step-wise Selection: Although the authors claim a net speedup, re-computing importance scores and sorting tokens at every decoding step introduces a new per-step overhead. While justified by the reduction in attention computation, a more detailed breakdown of this specific overhead would be beneficial.
2. Some important design choices are under-analyzed in the main paper. Since the method depends on the anchor layer and temperature scaling, I would have expected a more explicit sensitivity study for these choices, beyond the ablations on fixed vs. random vs. dynamic budgets. These design philosophy should be clearer elaborated.
3. The validation breadth is still somewhat limited relative to the scope of the claim. The motivation discusses both images and videos, but the reported experiments are centered on image-style multimodal reasoning benchmarks, with broader model-family evaluation shown only on a small number of additional backbones.

---

> ### Author Rebuttal · Authors · 2026-03-31
>
> **W1. Computational Overhead of Step-wise Selection.**
>
> **A:** Thank you for pointing this out. Step-wise visual selection introduces  additional overhead from anchor scoring and top-k selection. To remain compatible with FlashAttention, we adopt a block-level design, where the anchor layer outputs both the block-level importance scores and the updated hidden states in a single pass. We provide a kernel-level breakdown below:
>
> |Visual Context Length | Anchor scoring (ms)| Block selection (ms)| Sparse attention (ms)| Dense attention (ms) |
> |-|:-:|:-:|:-:|:-:|
> |8k|0.79|0.47|2.50|11.88|
> |16k|1.43|0.47|2.95|21.76|
> |32k|2.77|0.47|4.36|40.89|
>
> From this breakdown, we observe that the selection stage does lead to additional overhead. However, the subsequent computation benefits substantially from sparse visual attention, which makes the added selection overhead well justified. We will include this breakdown and clarify this effciency trade-off in the revision.
>
> **W2. Some important design choices are under-analyzed in the main paper.**
>
> **A:** Thanks. We will separately elaborate these design choices.
>
> **Temperature Scaling**
>
> As shown in Appendix A.2, a large number of inactive visual tokens induce a long-tail attention distribution, causing larger budget allocation. To mitigate this, we introduce temperature scaling to reduce long-tail noise while enabling controllable pruning ratios. We conduct an ablation study on RealWorldQA over different temperature values under the Retain $\leq$ 5% Tokens setting, as shown below:
>
> |Temperature $\tau$|0.02|0.05|0.1|0.4|0.7|1.0|Baseline|
> |-|:-:|:-:|:-:|:-:|:-:|:-:|:-:|
> |Avg Ret.(%)|1.89|1.95|2.08|3.15|4.86|4.98|100|
> |Avg Gen.|496.7|536.0|538.0|566.0|579.7|573.3|678.0|
> |Avg Acc.(%)|71.7|72.2|72.5|71.3|70.2|70.6|72.8|
>
> The results show a clear trade-off controlled by temperature scaling. Larger $\tau$ retains more tokens but yields longer generation without improving accuracy. A smaller $\tau$ over-sharpens the distribution and provides a lower compression ratio which may discard useful information. The best performance is at $\tau = 0.1$, achieving the best accuracy-pruning trade-off. Overall, these results support temperature scaling as an effective way to control pruning strength and align the visual budget with step-wise demand.
>
> **Anchor Layer Selection**
>
> We follow prior findings from PDrop [1] and Vscan [2], which show that visual dependency is stronger in earlier layers and becomes sparser in middle-to-late layers. We further observe that visual attention becomes stable from the middle layers onward (as shown in the [supplementary figure](https://anonymous.4open.science/r/ICML_2026_Rebuttal-2713/fig_sim.pdf)), indicating that reliable importance estimation can be obtained from this stage, which aligns with the conclusions drawn in PDrop and Vscan. Therefore, for FastV-VP, we choose the anchor layer at 1/2 of the total depth, i.e., the 17th layer. It is also worth noting that the anchor layer selection may vary w.r.t. the importance estimation method. For example, PDrop-VP follows the default three-stage setting of PDrop and selects anchor layers at 1/4, 1/2, and 3/4 of the total depth.
>
> **W3. The validation breadth is still somewhat limited relative to the scope of the claim.**
>
> **A:**  Thanks. VisionPulse dynamically determines the step-wise retention budget and performs visual token selection during reasoning. As these mechanisms are not specific to images, it naturally generalizes to other modalities (e.g., video) and across different LMM families.
>
> To address the concern on generality across modalities, we further evaluate VisionPulse on a randomly sampled 1/4 subset of VideoMME with 32 frames. As shown in the table below, VisionPulse still shows promising effectiveness, maintaining comparable accuracy while reducing generated tokens.
>
> |Qwen3-VL Thinking 4B|Gen.|Short|Medium|Long|Overall|
> |-|:-:|:-:|:-:|:-:|:-:|
> |Baseline|653.3|**74.7**|55.2|53.2|61.3|
> |FastV-VP (Retain$<=10$%Tokens) |507.5|72.2|55.7|**54.9**|61.2|
> |FastV-VP (Retain$<=5$%Tokens) |605.7|**74.7**|**56.2**|53.6|**61.8**|
>
>
> To address the concern on model-family breadth, we further expand the backbone evaluation to InternVL-3.5 8B (Retain $\leq$ 5% tokens) on three datasets: MMVet, RealWorldQA, and MIA-Bench. The average performance across three datasets is shown in the table below. We find that VisionPulse consistently maintains competitive performance while reducing generation length. We will include more video benchmarks and LMMs in the future revision.
>
> |Method|Avg Gen.|Avg Acc.|
> |-|:-:|:-:|
> | Baseline|1770.3|68.2|
> | FastV|2586.1|55.8|
> | FastV-VP|1587.2|68.0|
>
> [1] PyramidDrop: Accelerating Your Large Vision-Language Models via Pyramid Visual Redundancy Reduction. CVPR 2025.\
> [2] VScan: Rethinking Visual Token Reduction for Efficient Large Vision-Language Models. TMLR 2026.

---

> > ### Author Rebuttal · Reviewer_XTDy · 2026-04-01
> >
> > While the method shows strong results in the image domain, the performance gains in the video setting are notably less significant. Could the authors provide further insights or analysis into why this discrepancy exists?

---

> > > ### Author Response · Authors · 2026-04-02
> > >
> > > Thank you once again for taking the time to review our paper and for providing such insightful and constructive feedback.
> > >
> > > In terms of the overall accuracy, VisionPulse still maintains comparable results and even yields improvements in the video setting, which is actually stronger than our observations in the image settings. Meanwhile, we also report the overall speedup of two retention settings in the table below. As shown in this table, under the 5% and 10% retention settings, VisionPulse can achieve a 1.25× and 1.44× speedup, respectively. These results suggest that VisionPulse remains effective and efficient in the video setting. When the two VisionPulse  methods (with different retention settings) are compared to each other, we think they have comparable comprehensive performance by taking a trade-off between the overall accuracy and speedup.
> > >
> > > | Qwen3-VL Thinking 4B | Gen. $\downarrow$ | Short | Medium | Long | Acc. $\uparrow$ | Speedup $\uparrow$ |
> > > |-|:-|:-:|:-:|:-:|:-:|:-:|
> > > |Baseline|653.3|**74.7**|55.2|53.2|61.3||
> > > |FastV-VP(Retain$<=10$%Tokens)|507.5|72.2|55.7|**54.9**|61.2|**1.44$\times$**|
> > > |FastV-VP(Retain$<=5$%Tokens)|605.7|**74.7**|**56.2**|53.6|**61.8**|1.25$\times$|

---

### Decision · Program_Chairs · 2026-04-30

**Decision:**

Accept (regular)

**Comment:**

The submission initially received mixed reviews. The main concerns about the submissions are: 1) More comprehensive evaluations on video and various backbones are needed. 2) More ablation studies (such as explicit sensitivity studies) are expected. 3)  In the presentation, the paper does not clearly compare the performance between training-free dynamic pruning and training-time sparsification, and the explanation of the core advantage of the training-free design is not in-depth enough. 4) There is no clear explanation of the unique advantages of the practical value of the training-free design. 5) More detailed comparisons with other methods (such as prefilling token reduction methods) will demonstrate its effectiveness. 6) It is reasonable to extend the proposed method to MLLM. After the rebuttal, many concerns are addressed. The remaining concerns are: Concern 6 (extension to MLLM) and Concern 4, where the reviewer czjy didn't respond to the rebuttal.

Overall, I think the submission is a high-quality submission, and I recommend **Weak Accept**.  And the authors need to incorporate all these new results in the rebuttal to the final version.